# Structural basis for antibiotic transport and inhibition in PepT2

Joanne L. Parker [1,2,4] ✉, Justin C. Deme [3,4], Simon M. Lichtinger [1,4], Gabriel Kuteyi[1], Philip C. Biggin [1] ✉, Susan M. Lea [3] ✉ & Simon Newstead [1,2] ✉

The uptake and elimination of beta-lactam antibiotics in the human body are facilitated by the proton-coupled peptide transporters PepT1 (SLC15A1) and PepT2 (SLC15A2). The mechanism by which SLC15 family transporters recognize and discriminate between different drug classes and dietary peptides remains unclear, hampering efforts to improve antibiotic pharmacokinetics through targeted drug design and delivery. Here, we present cryo-EM structures of the proton-coupled peptide transporter, PepT2 from *Rattus norvegicus*, in complex with the widely used beta-lactam antibiotics cefadroxil, amoxicillin and cloxacillin. Our structures, combined with pharmacophore mapping, molecular dynamics simulations and biochemical assays, establish the mechanism of beta-lactam antibiotic recognition and the important role of protonation in drug binding and transport.

Antibiotics are a vital part of modern healthcare systems and one of the most significant advances in modern medicine[1]. Beta-lactams are among the most widely prescribed and effective antibiotics, with a broad range of activity against most pathogenic bacteria[2]. Beta-lactams function by inhibiting the synthesis of the peptidoglycan layer in the bacterial cell wall[3]. Structurally, the beta-lactam antibiotics are currently separated into four classes: penicillins, cephalosporins, carbapenems and monobactams, which differ in either the R groups attached to either the four-membered beta-lactam ring or in the case of the penicillin, carbapenem and cephalosporins, the attached five or six-membered rings[4]. Due to the wide range of derivatives based on these core scaffolds, the uptake and retention of beta-lactam antibiotics within the human body vary partly due to their specific interactions with different families of solute carrier (SLC) transporters[5–7]. A particular focus in current antibiotic drug development is improving the oral bioavailability of carbapenems, which are highly effective against both Gram-negative, and Gram-positive bacteria. However, current formulations of carbapenems show increased breakdown in the gut and poor transport across the gut epithelia, restricting the options available to clinicians in delivering these drugs to patients[8]. Understanding the interactions between small molecule drugs and solute carriers is a promising route to improving drug pharmacokinetics and efficacy[9].

The proton-coupled peptide transporters PepT1 (SLC15A1) and PepT2 (SLC15A2) have been extensively studied due to their influence on the oral bioavailability and renal clearance of beta-lactam antibiotics in the body[7,10–21]. The primary physiological role of the SLC15 family is the absorption and retention of dietary nitrogen in the form of di- and tripeptides[22,23]. Ingested protein is broken down and transported across the intestinal brush border membrane via the plasma membrane peptide transporter, PepT1[24–26]. In contrast, circulating peptides are retained in the body through reabsorption via PepT2, which selectively retains peptides in the kidney[27,28], and also regulates peptide transport across the blood-brain barrier[7,29] (Fig. 1a). PepT1 and PepT2 are unusual solute carriers in being highly promiscuous[30], able to recognize a large and diverse range of chemically distinct substrates[31]. The ability to recognize different chemical groups underpins the role played by these proteins in beta-lactam antibiotic uptake, as these drugs display both steric and chemical similarity to tripeptides[32].

PepT1 and PepT2 are members of the Proton-coupled Oligopeptide Transporter or POT family, which is widely distributed within pro- and eukaryotic genomes[30]. POT family transporters belong to the

[1]Department of Biochemistry, University of Oxford, Oxford, UK. [2]The Kavli Institute for Nanoscience Discovery, University of Oxford, Oxford, UK. [3]Center for Structural Biology, Center for Cancer Research, National Cancer Institute, Frederick, USA. [4]These authors contributed equally: Joanne L. Parker, Justin C. Deme, Simon M. Lichtinger. ✉e-mail: joanne.parker@bioch.ox.ac.uk; philip.biggin@bioch.ox.ac.uk; susan.lea@nih.gov; simon.newstead@bioch.ox.ac.uk

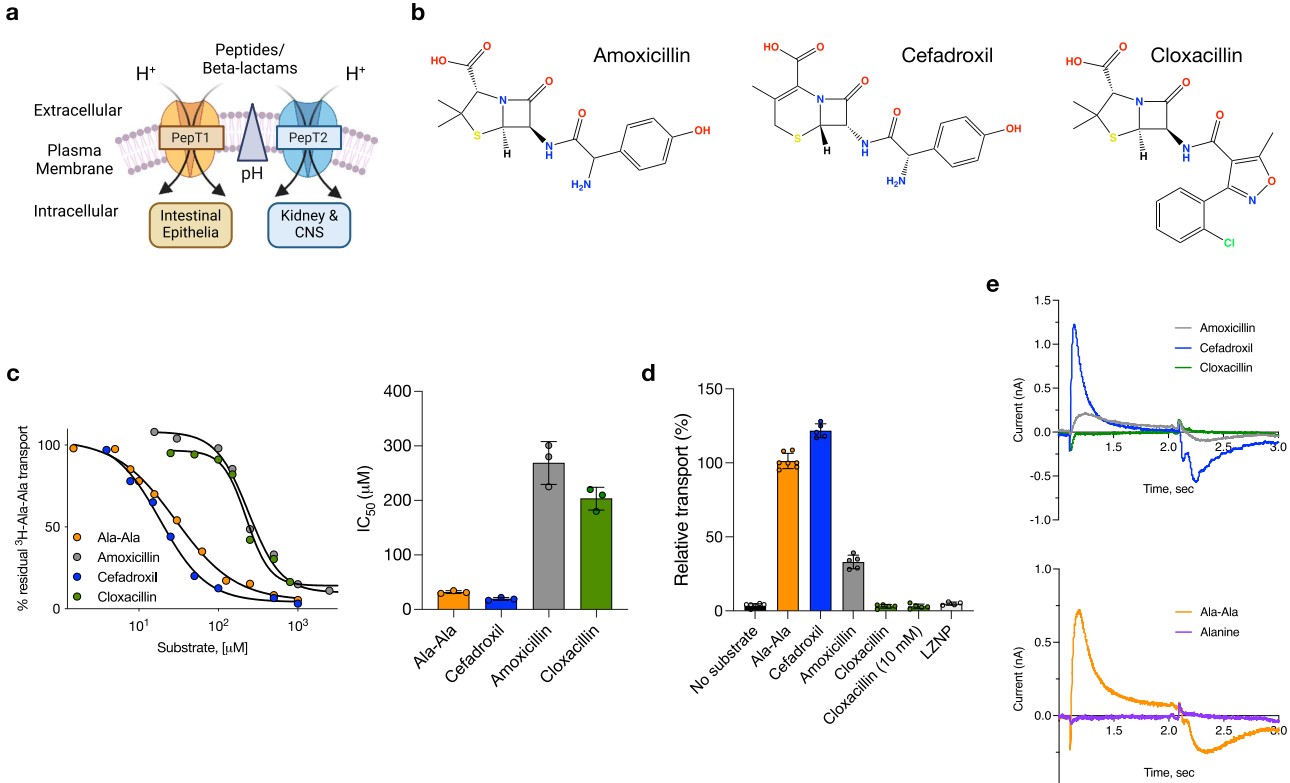

**Fig. 1 | Functional characterization of beta-lactam transport by PepT2.**
**a** Overview of PepT1 and PepT2 function in the body. Peptides are transported into the cell via PepT1 and PepT2, driven by the inwardly directed proton gradient $\Delta\mu H^+$ (acidic outside). **b** Chemical structures of cefadroxil, amoxicillin and cloxacillin. **c** Representative $IC_{50}$ data for the antibiotics and di-alanine peptide ($n = 3$ biological experiments performed on different days, mean and errors shown are SD). **d** Counterflow assay data showing the ability of only cefadroxil and amoxicillin to

drive transport. LZNP refers to the inhibitor of PepT1 & 2, Lys [Z(NO$_2$)]-Proline. 0.5 mM of each ligand was used unless stated otherwise. ($n = 5$ independent experiments performed on different days, mean shown and errors indicate SD). **e** Representative solid support membrane recordings for the transport of di-alanine peptide, cefadroxil and amoxicillin and the non-transported compounds, alanine and cloxacillin. Created in BioRender. Newstead, S. (2023) BioRender.com/f58m033.

Major Facilitator Superfamily (MFS). They are all proton (H$^+$) driven symporters, using the inwardly direct proton electrochemical gradient ($\Delta\mu H^+$) to drive the concentrative uptake of peptides and drugs into the cell[32]. Recently, the structures of PepT1 and PepT2 have been reported from both human and mammalian species, providing valuable insights into the mechanisms underpinning peptide and prodrug recognition[33–35]. These studies build on previous structural and biochemical studies on the prokaryotic POT family members to understand the structural basis for ligand promiscuity[36–55]. However, the molecular basis for the selective recognition and uptake of beta-lactam antibiotics remains unclear, hampering efforts to improve the pharmacokinetic profiles of new antibiotic drugs.

To address this important aspect of SLC15 function, we determine the structure of PepT2 from *Rattus norvegicus* in complex with three different beta-lactam antibiotics. Using in vitro assays, we demonstrate that one of the beta-lactam antibiotics, cloxacillin, functions as a competitive inhibitor. Combining our cryo-EM structures with molecular dynamics to probe the role of protonation in substrate recognition, we identify a crucial role for the primary amine group in orienting beta-lactam antibiotics in the binding site and proton binding in locking the drug via the carboxylate group. Our results establish a working pharmacophore model for beta-lactam recognition and explain the differences between substrate binding and inhibition mechanisms.

## Results

### Characterization of beta-lactam transport

The proton-coupled peptide transporters are known to transport several families of beta-lactam antibiotics, including cephalosporins and penicillins. Although the substrate range recognized is extensive, encompassing many different drug classes, substrate specificity exists within the beta-lactam families (Fig. 1b). Some antibiotics exhibit high affinity, such as cefadroxil with a $K_i$ 3 μM, while others show medium affinity, such as amoxicillin with a $K_i$ 0.2–0.43 mM and low affinity, such as cloxacillin $K_i$ 1 mM[10,19]. However, to date, the transport has been primarily characterized using studies reporting the inhibition of a radioactive reporter peptide in cell-based assays[10,56]. To verify the $K_i$ values reported for these cell-based studies, we reconstituted rat PepT2 into liposomes and calculated $IC_{50}$ values using inhibition of the uptake of radioactive di-alanine peptide (Fig. 1c). While cefadroxil displayed an $IC_{50}$ of 19 μM ± 3, both amoxicillin and cloxacillin displayed significantly weaker values of 270 μM ± 39 and 203 μM ± 21, respectively. In comparison, the di-alanine peptide had an $IC_{50}$ of 32 μM ± 3. However, a drawback of $IC_{50}$ studies is the inability to discriminate between transported ligands and non-transported inhibitors. Therefore, we tested the ability of these antibiotics to drive transport via a counterflow experiment compared to the di-alanine peptide (Fig. 1d). As expected, cefadroxil was able to drive transport more effectively than di-alanine, consistent with the calculated $IC_{50}$ values.

Similarly, amoxicillin was less effective, showing ~ 35% uptake. However, cloxacillin was unable to drive transport, even at high concentrations (10 mM). To verify the inhibitory property of this antibiotic, we used the negative control of Lys[Z(NO$_2$)]-Pro (LZNP), a known high-affinity inhibitor of PepT2[57]. Recently, the use of solid support membrane devices has enabled measurements of changes in membrane capacitance following charge movement via secondary active transporters[58]. We analysed the activity of the reconstituted

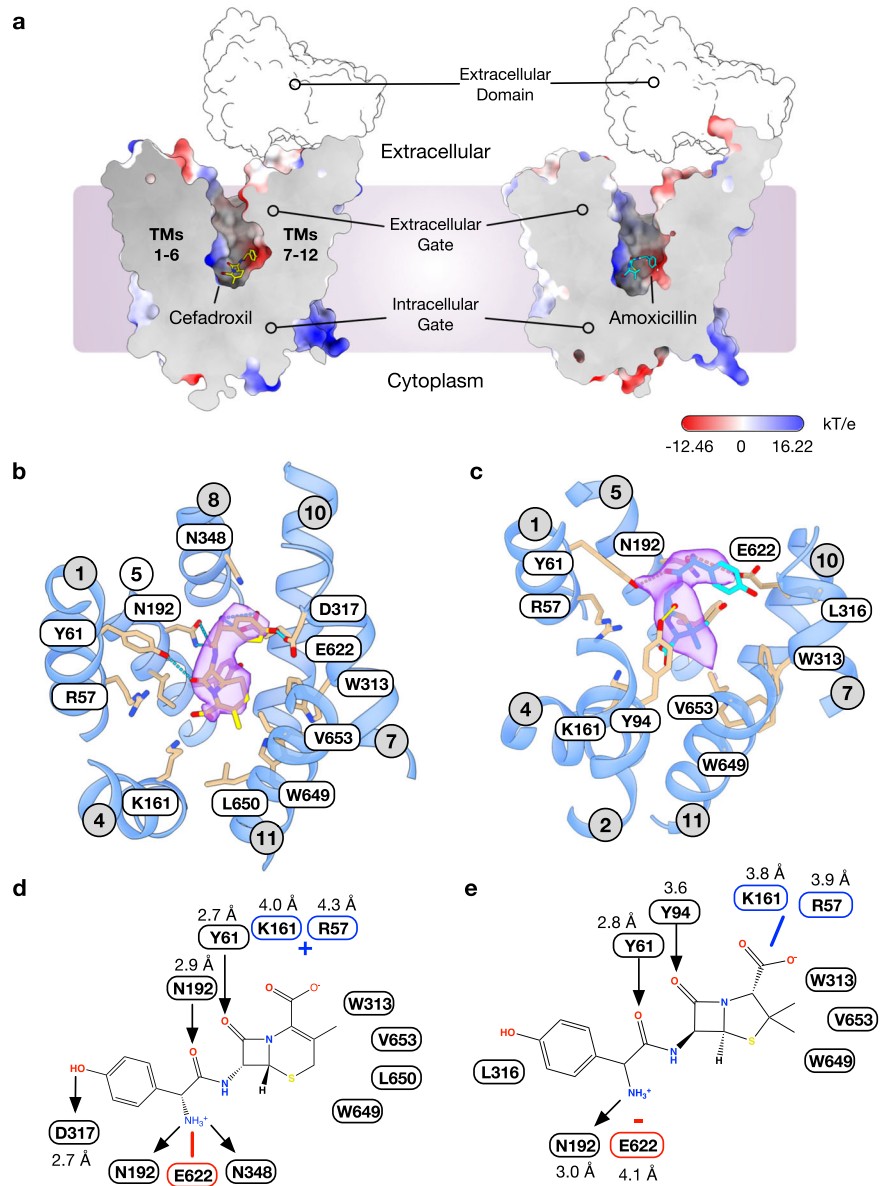

**Fig. 2 | Cryo-EM structure of antibiotic-PepT2 complexes. a** Electrostatic surface representation of the cryo-EM structure of PepT2 bound to cefadroxil and amoxicillin, highlighting the key structural features of the transporter. Cefadroxil and amoxicillin are shown in a stick representation. The position of the extracellular domain is indicated but not coloured due to it being absent in the deposited models. **b** The binding site of PepT2 shows the bound cefadroxil antibiotic (yellow sticks) with nearby and interacting side chains. The cryo-EM density is shown in purple, contoured at a threshold of 0.487. Hydrogen bonds are indicated (cyan dashed lines). Side chains and transmembrane helices are labelled. **c** The binding site of PepT2 shows the bound amoxicillin (cyan sticks). The cryo-EM density is shown in purple, contoured at a threshold of 0.442. Hydrogen bonds are indicated (yellow dashed lines). **d** Schematic showing the interactions between PepT2 and cefadroxil. Hydrogen bond donors and acceptors are indicated by arrows, and coloured lines indicate electrostatic interactions. Where electrostatic distances exceed 4 Å the interactions are indicated by positive and negative signs. Distances (Å) are calculated from heavy atoms. **e** Schematic interaction map between PepT2 and amoxicillin.

PepT2 using the SURFE²R platform, which also confirmed the radioactive uptake and counterflow assays (Fig. 1e). Whilst the addition of di-alanine, cefadroxil and amoxicillin all generated a significant capacitance change, the addition of cloxacillin did not. Thus, we conclude that cloxacillin is not transported by PepT2 but likely acts as a competitive inhibitor. We considered the question of why certain antibiotics function as ligands while others are inhibitors, as this behaviour has implications for inhibitor design more broadly within the SLC15 family, which includes members linked to inflammatory regulation[59]. We, therefore, sought to understand the nature of the binding interactions that determine good compared to weak substrates and verify the inhibitory mechanism of cloxacillin.

## Cryo-EM structure of PepT2 in complex with Cefadroxil and Amoxicillin

To gain further insight into the mechanism of antibiotic recognition, we determined the structure of PepT2 in complex with both cefadroxil and amoxicillin at 3.1 Å and 3.2 Å, respectively, using cryo-EM (Fig. 2a–c, Supplementary Table 1, Supplementary Figs. 1 and 2). The structures were obtained using a nanobody identified in our previous study reporting the apo structure[33]. In both structures, the transporter adopts an outward open, extracellular facing state, with a large solvent-accessible cavity extending from the exterior of the transporter down towards the intracellular gate, which is constructed from the packing together of TMs 4–5 and TMs 10–11 (Supplementary Fig. 3).

The overall arrangement of TM helices in both structures is essentially identical to the previously obtained apo structure of rat PepT2 (PDB:7NQK), except the Coulomb density map around the extracellular domain (ECD) is much lower resolution, resulting in our decision not to build the IgG domains in these structures. However, clear density for the nanobody was obtained, which binds to the extracellular regions of TMs 1 and 2[33] (Supplementary Figs. 1 and 2). The structures have root mean square deviations (r.m.s.d) of 0.62 Å and 0.45 Å for the cefadroxil and amoxicillin structures, respectively, when aligned against the TM helices of the apo state of the rat PepT2 (PDB:7NQK). When compared to the L-Ala-L-Phe peptide bound state of human PepT1 (PDB:7PMX), which was captured in the same outward open conformation[34], the antibiotic bound structures have an r.m.s.d of 1.18 Å and 1.19 Å when aligned against the TM helices of the dipeptide bound PepT1 transporter.

Cefadroxil can be clearly observed in the map and adopts an elongated 'L-shaped' conformation (Fig. 2b, d and Supplementary Fig. 4). The hydroxyphenyl group interacts with Asp317 on the extracellular gating helix, TM7, which also forms part of a negatively charged pocket within the binding site[33]. The amino group sits adjacent to Glu622 on the intracellular gating helix, TM10, and also interacts with Asn192 on TM4, which forms part of the intracellular gating helices in the N-terminal bundle and Asn348 on TM8, which along with TM7 forms part of the extracellular gate within MFS transporters[60]. The carbonyl group of cefadroxil also interacts with Asn192, anchoring this region of the drug within the transporter. The remainder of the drug molecule makes fewer direct interactions with the binding site; the carbonyl group in the beta-lactam ring hydrogen bonds to Tyr61 on the extracellular gating helix, TM1, while the methyl group at the C3 position on the dihydrothiazine ring sits within a hydrophobic pocket, observed previously in bacterial homologues of the POT family[37], and formed by Leu650, Val653 and Trp649 on the intracellular gating helix TM11. The carboxyl group sits within the positively charged pocket formed by Arg57 on TM1 and Lys161 on TM4, effectively clamping the drug within the binding site between two opposing electrostatic charges. Previous structures of peptide-bound prokaryotic POT family homologues and human PepT2 have identified the critical role electrostatics play in correctly orientating peptides within the binding site[34,37,44]. Specifically, the conserved glutamate, Glu622 on TM10 and Arg57 on TM1, function to clamp the amino and carboxy termini of peptides, respectively.

Amoxicillin adopts a similar binding position to that observed for cefadroxil, with the hydroxyphenyl, amino and carboxylate groups sitting in equivalent positions in the binding site (Fig. 2c, e and Supplementary Fig. 4). However, the beta-lactam ring is rotated ~90° relative to cefadroxil, with the carbonyl group pointing towards the extracellular entrance to the binding pocket. The two methyl groups on the 5-membered thiazolidine ring point towards the hydrophobic pocket near TM11 and side chains Trp313, Val653 and Trp649. Consistent with the lower IC$_{50}$ values, amoxicillin makes fewer direct interactions with the transporter, which appear to result from the altered orientation of the drug, presumably to accommodate the geometry of the thiazolidine ring. As observed in the cefadroxil structure, the amino group sits in the negatively charged pocket and interacts with Asn192 via a hydrogen bond but now sits a little further from Glu622 (~4.0 vs 2.4 Å). Tyr61 now interacts with the carbonyl group of the peptide bond, as opposed to the carbonyl on the beta-lactam ring observed in cefadroxil. As noted above, repositioning the beta-lactam ring substantially changes the location of the carbonyl group on the beta-lactam ring, which now interacts with Tyr94 on TM2. However, similar to cefadroxil the drug is positioned within the electrostatic clamp with the carboxyl group positioned close to Arg57 and Lys161. Together, these structures establish the important features driving the recognition of beta-lactam antibiotics within PepT2 and enable the comparison with physiological peptide binding, as discussed below.

## Structural basis for inhibition by Cloxacillin

Having established how PepT2 discriminates between a cephalosporin and an aminopenicillin, we next sought to uncover the mechanism of inhibition observed for cloxacillin, a semisynthetic penicillin carrying a 3-(2-chlorophenyl)−5-methylisoxazole-4-carboxamido group at position 6 of the beta-lactam ring. From the cryo-EM data, we could generate two maps derived from independent 3D classification schemes of a consensus particle set, which show two different positions for the drug within the binding site (Fig. 3a, Supplementary Figs. 5 and 6). Pose one was obtained from 106,684 particles and generated a map at 3.1 Å resolution (map 1), whereas pose two was obtained from 201,206 particles, which generated a map at 2.9 Å resolution (map 2) (Supplementary Table 1).

Similar to the cefadroxil and amoxicillin complex structures, the protein backbone showed no obvious differences to those obtained for the apo PepT2 (r.m.s.d of 0.35 Å for 531 C$_\alpha$ atoms). In pose 1, cloxacillin adopts a vertical orientation with the chlorophenyl group sitting within 3 Å of the Arg57 and the E$^{53}$xxER motif on TM1 and adjacent to the intracellular gate occlusion formed by TMs 4, 5 and 10, 11 packing against one another (Fig. 3b and Supplementary Figs. 3 and 6). The methylisoxazole group sits close to the hydrophobic pocket formed by Trp313, Trp649 and Tyr188, with the carbonyl group of the peptide bond sitting near (~4 Å) to Lys161. The beta-lactam ring makes very few interactions to side chains in the binding site; the sulphur atom in the thiazolidine ring sits close to Tyr61 and may make a weak hydrogen bond to the hydroxyl. At the end of the molecule, the carboxyl group hydrogen bonds with Tyr94 on TM1, which presumably stabilizes the vertical orientation in the binding site, while the beta-lactam ring carbonyl sits near to Tyr86. In pose 2, the chlorophenyl group adopts a similar location to pose 1, at the base of the binding site and sitting close to Glu53 in the E$^{53}$xxER motif, but now positioned closer to Ile191 and Arg57 (Fig. 3c and Supplementary Fig. 6). Similarly, the methyl group in the methylisoxazole ring extends towards Trp313 and Val653 in the hydrophobic pocket. However, the two poses differ more substantially in the location of the beta-lactam and thiazolidine rings. Whereas in pose 1, the drug molecule sits in roughly the centre of the binding site, in pose 2, cloxacillin sits to one side in an asymmetric position. Specifically, the peptide carbonyl group interacts with Tyr188 (TM5). Tyrosine 188 also interacts with the carbonyl group in the beta-lactam ring, while the sulphur atom interacts with Tyr61, similar to that observed in pose 1. Of note is the beta-lactam ring carbonyl, which in this pose now interacts with Glu622 (TM10) similar to the free amino group in cefadroxil and amoxicillin. However, we can model two rotamer positions for Glu622 in the maps, suggesting this interaction is not stable. The carboxyl group makes further hydrogen bond interactions with Asn192 (TM5), which also interacts with the nitrogen in the beta-lactam ring to complete the interaction map. Poses 1 and 2 for cloxacillin are however likely just two of several possible positions for this compound in the binding site.

Taken together, the two poses suggest that cloxacillin binding is dominated by the positioning of the chlorophenyl group at the base of the transporter, with no strict specificity for the location of the beta-lactam backbone. The observed inhibitory properties of cloxacillin likely result from its inability to adopt a stable binding pose. As discussed below, this inability is likely to result from the absence of a free amino group, which precludes the drug from engaging the electrostatic clamp formed between Arg57 and Glu622, which enables substrates to coordinate movement between the N- and C-terminal bundles.

## Interplay of protonation and ligand recognition

PepT2 is a proton-coupled transporter, and therefore, understanding how drugs interact with the binding site requires consideration of the protonation states of key protonatable side chains[33,36,40,61]. To gain further insights into substrate discrimination, proton coupling and

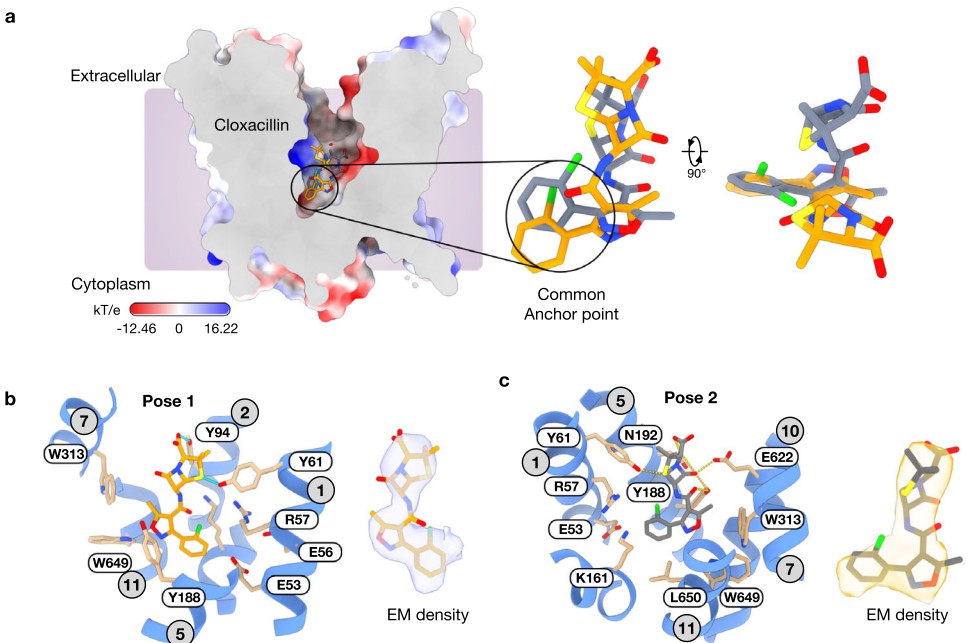

**Fig. 3 | The cloxacillin-bound structure of PepT2 reveals an inhibition mechanism. a** Electrostatic surface representation of the cryo-EM structure of PepT2 bound to cloxacillin, showing the two orientations overlaid in the binding site. **b** Zoomed view of the PepT2 binding site, showing the orientation and interactions for pose 1. Side chains and transmembrane helices are labelled. **c** Pose 2. The cryo-EM density for each pose is shown on the right, contoured at a threshold level - 0.22. Side chains and transmembrane helices are labelled.

inhibition in PepT2, we undertook unbiased molecular dynamics (MD) simulations of the drug molecules using the cryo-EM structures as starting poses (Fig. 4a and Supplementary Fig. 7). For both cefadroxil and amoxicillin, three sets of simulations were run with the protonation state of Glu53 and Glu56 modulated (see methods for details and consideration of pKa values used). When both Glu53 and Glu56 are deprotonated, i.e. in their standard protonation states at pH 7.5, the amino group of cefadroxil remains stably bound to Glu622, and with some flexibility also to Asn192 and Asn348 (Supplementary Fig. 8). In contrast, the carboxylate group is unstable and sits ~6–10 Å away from Arg57 (Fig. 4a). These results are consistent with the interaction network observed in the cryo-EM structure (Fig. 2) and previous MD simulations of peptide ligands, which highlighted the role of the amino terminus in the initial capture of ligands[33]. However, following the protonation of Glu56 and, to a lesser extent, Glu53, the carboxylate group of cefadroxil moves to interact with Arg57 (Fig. 4b). The same pattern of interactions occurs in the amoxicillin simulations, with the protonation of Glu56 releasing Arg57 to clamp the carboxylate group. However, when Glu56 is protonated, and to a lesser extent with Glu53 protonated, the amino group of amoxicillin moves away from Glu622 and detaches entirely from Asn192 (Supplementary Fig. 8). This change to the orientation can be visualized by projecting the pooled trajectories onto a 2D plane, representing a slice through the 3D volume of the binding site (Fig. 4c). These 2D plots illustrate the extent to which protonation of Glu56 moves the centre of mass for the carboxylate group of both cefadroxil and amoxicillin closer to Arg57, which establishes the necessary interactions between the ligand and N- and C-terminal bundles of the transporter. The observation that amoxicillin cannot interact stably with the Glu622, Asn192 and Asn348 triad at the amino group and Arg57 at the carboxyl group simultaneously provides a reasonable explanation for its reduced experimental affinity compared to cefadroxil (Fig. 1c).

We next validated the significance of our results from the unbiased MD by running absolute binding free energy (ABFE) simulations of cefadroxil and amoxicillin for deprotonated and protonated Glu56 (see methods). The results indicate that cefadroxil and amoxicillin have similar binding free energies to the transporter in their cryo-EM poses (Table 1). Once Glu56 is protonated, however, cefadroxil experiences a substantial gain in affinity of ~6 kcal mol⁻¹, while amoxicillin affinity is reduced by ~1.7 kcal mol⁻¹, supporting the conclusions drawn from the unbiased MD runs above.

The cryo-EM structures of cloxacillin revealed a more dynamic binding mode compared to either cefadroxil or amoxicillin (Fig. 3). Indeed, our MD simulations support these observations, as cloxacillin fails to adopt a stable binding pose within the 6 × 1 μs trajectories for each of the two cryo-EM models (Supplementary Fig. 9). The absence of an amino group results in no significant interaction with Glu622 by any functional group in the drug molecule. Similarly, the carboxylate group also fails to interact with Arg57 with increased frequency following Glu56 protonation. The failure of cloxacillin to engage in the specific binding pocket interactions formed by cefadroxil and amoxicillin is also quantitatively reflected in ABFE affinities (Table 1). When Glu56 is deprotonated, the affinity is ~4 kcal mol⁻¹ lower than either cefadroxil or amoxicillin, and unlike the two transported antibiotics, there is no stabilization of binding upon Glu56 protonation (Supplementary Fig. 9). As discussed below, the likely mechanism for inhibition appears to be simple steric occlusion and failure to trigger the necessary interactions to Glu622 and Arg57 required for transport. It is interesting to note that the MD does not pick out the two cryo-EM poses as being particularly stable. Generating accurate force-fields for drug molecules still represents a significant challenge due to the large number of possible atom combinations[62]. The discrepancy between the MD for cloxacillin and the cryo-EM data is likely to reflect limitations in the force-field, which are amplified by the weak binding properties of this drug. Nevertheless, the inability of the MD to replicate the cryo-EM poses is qualitatively consistent with the inability of cloxacillin to engage strongly with the transporter.

## Structural discrimination between substrates and inhibitors within the beta-lactam family

The inhibitory property of cloxacillin was unexpected but presented an opportunity to decode further the structural differences between

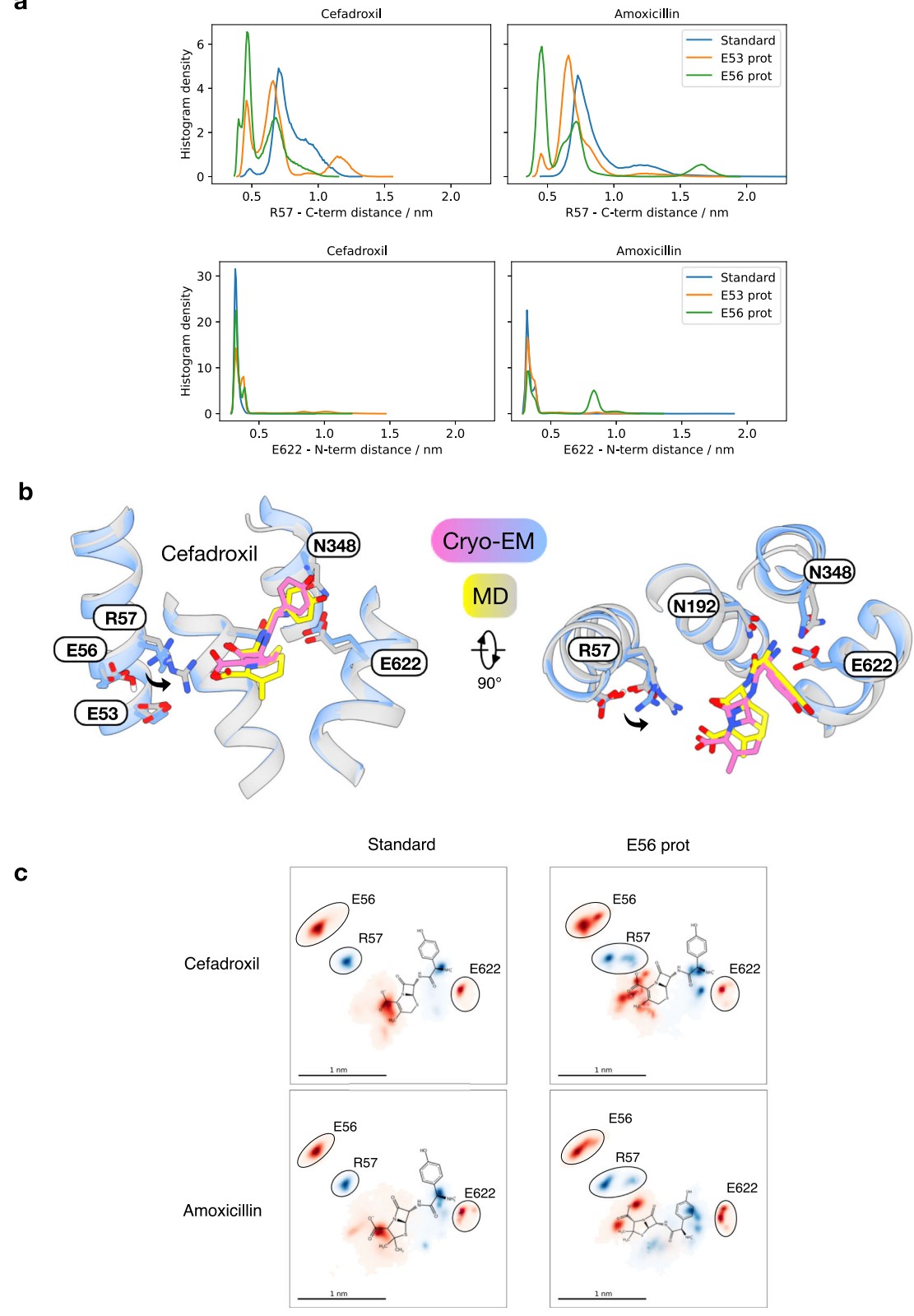

**Fig. 4 | Protonation of Glu56 promotes ligand recognition via the E$^{53}$xxER motif. a** Microsecond-long unbiased molecular dynamics (MD) simulations starting at the cefadroxil and amoxicillin cryo-EM models, using 6 replicates for each condition (standard protonation state, E53 protonated and E56 protonated). Histograms of the pooled trajectories of each condition are shown for the drug N-terminus (amino N) – E622 (Cδ) distance (upper block) and drug carboxyl carbon – R57 (Cζ) distance (lower block). **b** Structural overlay of an example frame from a 1μs-long replicate taken in the last 200 ns with the cryo-EM structure of the cefadroxil complex. **c** Pooled replicate trajectories (6 μs total, same trajectories as in part a, cefadroxil and amoxicillin standard protonation states and E56 protonated, respectively) projected onto the plane spanned by the Cα atoms of E622, R57 and W313. Positions of E622, R57, the drug N- and C-terminus (all as defined above) and E56 (Cδ) are shown as 2D-histograms in the plane. Colours correspond to the different charges shown (red – negative; blue – positive). Protein residue densities were surrounded with black ovals, with the chemical structures of cefadroxil and amoxicillin overlaid for illustration.

substrates and inhibitors within the beta-lactam family[63]. For a comparative analysis, we tested five additional antibiotics from the penicillin and cephalosporin families. Using our counterflow assay, we first determined that PepT2 cannot transport either moxalactam, ceftibuten or benzylpenicillin (Fig. 5a).

Cefaclor, however, acts as a good substrate, achieving 76% activity relative to the physiological di-alanine peptide in the counterflow assay, and an $IC_{50}$ of 60 μM (Fig. 5b). Ampicillin is also transported, albeit at a lower level than cefaclor, achieving 13% activity in the counterflow assay and an $IC_{50}$ of 750 μM. This compares to $IC_{50}$ value for moxalactam of 500 μM (Supplementary Table 2). Together with our previous analysis of cefadroxil, amoxicillin and cloxacillin, we can now separate these beta-lactam drugs into substrates and inhibitors (Fig. 5c). Of note is the hydroxyl on the hydroxyphenyl group of cefadroxil and amoxicillin, which is absent in cefaclor and ampicillin respectively. In both cases, cefadroxil and amoxicillin are better substrates for PepT2 than either cefaclor or ampicillin, considering their $IC_{50}$ values and counterflow behaviour (Supplementary Table 2). A plausible explanation for the increased affinity comes from the cryo-EM structure of cefadroxil, which reveals the hydroxyl is positioned close to a key gating helix, TM7 and interacts directly with Asp317

(Fig. 2b,d). Aspartate 317 forms part of an interaction network that controls the extracellular gate dynamics in response to proton binding to a conserved histidine on TM2, His87 in PepT2[36,64]. Thus, the interaction between the ring hydroxyl in cefadroxil and amoxicillin with Asp317 positively affects both the recognition and transport of these drugs via PepT2.

Conversely, when assessing the inhibitory potency of the beta-lactam drugs, our data shows that cloxacillin, with an $IC_{50}$ of 203 μM, is more effective than moxalactam (500 μM), benzylpenicillin (1 mM) or ceftibuten (1 mM). Our MD analysis of cloxacillin shows the drug is unable to adopt a stable binding orientation with respect to Glu622 or Arg57 (Supplementary Fig. 9), which is consistent with the two binding poses we observe in the cryo-EM structures (Fig. 3). A notable difference between cloxacillin and amoxicillin is the replacement of the primary amine group in the former compound for a methylisoxazole group in the latter. The absence of the primary amine removes a positive charge from cloxacillin and, therefore, the ability of the antibiotic to engage the Glu622, Asn192 and Asn348 triad on TMs 10, 4 and 9, respectively. Without this anchoring interaction, the binding of the drug is dominated by the chlorophenyl moiety, which in the cryo-EM structures binds in a pocket below Arg57 (Fig. 3). Interestingly, the chlorophenyl moiety of cloxacillin sits in a similar position to that modelled previously for the prodrug valacyclovir[33], indicating that the occupation of this pocket is not the reason for the inability of cloxacillin to trigger transport. More likely, the failure to adopt a stable binding pose and stably engage Glu622 explains the inhibitory properties of this drug. It is also consistent with the requirement of a free amino terminus for peptide ligands[27,65]. Neither moxalactam nor benzylpenicillin have primary amine groups, so the most likely explanation for their ability to inhibit PepT2 is their inability to stably engage Glu622, as observed for cloxacillin. Our results from ceftibuten, which does contain a primary amine, also highlight the importance of distance between the free amino and carboxyl groups in beta-lactam substrates, which was previously demonstrated for orally active prodrugs[66]. The addition of the amino-thiazol group extends the primary amine at a distance equivalent to a tetrapeptide, which is similar to moxalactam and likely makes these drugs too large to transport.

**Table 1 | Impact of protonation on the binding free energies of selected antibiotics in PepT2**

| Ligand | Glu56 | ΔG / kcal mol$^{-1}$ |
|---|---|---|
| Cefadroxil | not protonated | −12.0 ± 1.5 |
| | protonated | −18.0 ± 0.5 |
| Amoxicillin | not protonated | −10.6 ± 1.7 |
| | protonated | −8.9 ± 1.1 |
| Cloxacillin | not protonated | ~ −6.8* |
| | protonated | ~ −5.5* |

Absolute Binding Free Energy values were obtained (see methods for details) with either Glu56 protonated or deprotonated for the antibiotic complexes. Data reported is the mean (from 5 replicates) and errors are SD. *Note that for cloxacillin, as there was no binding pose stable on the equilibration timescale, we could not derive replicates for error bars in the same way as for cefadroxil and amoxicillin.

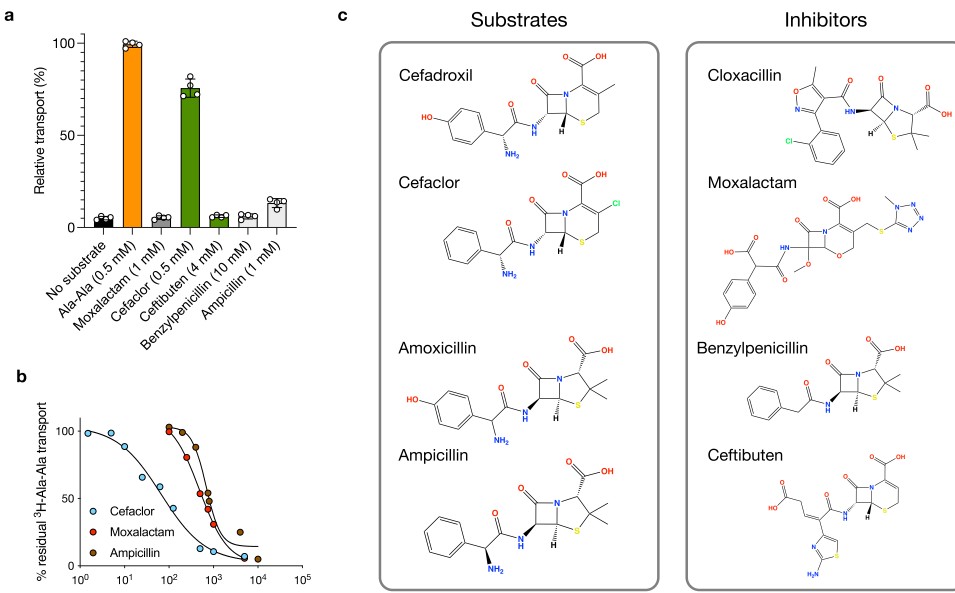

**Fig. 5 | Interaction of different beta-lactam antibiotics with PepT2.**
**a** Counterflow transport assays to discriminate substrates from inhibitors. ($n = 4$ independent biological experiments performed on different days and mean shown and errors indicate SD). **b** Representative $IC_{50}$ data for the antibiotics cefaclor, moxalactam and ampicillin. **c** Classification of the tested beta-lactam antibiotics into substrates or inhibitors.

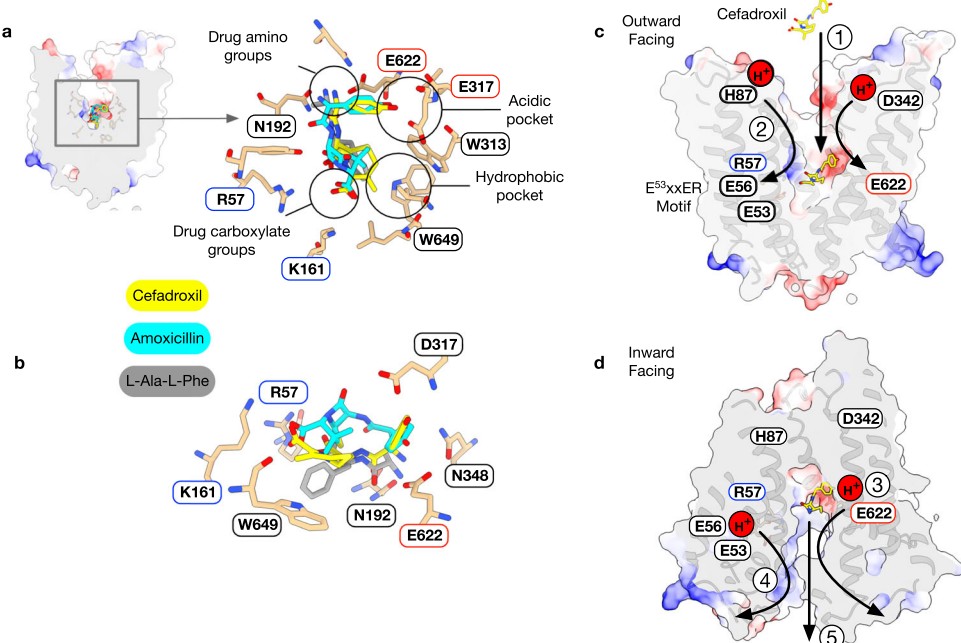

**Fig. 6 | Pharmacophore model and transport mechanism for beta-lactam antibiotics. a** Overlay of the cefadroxil and amoxicillin structures (this study) with the peptide-bound structure of human PepT1[54] (PDB:7PMX). Conserved anchor points for the substrate amino and carboxylate groups are shown with conserved pockets within the transporter binding site. **b** horizontal view of the binding site illustrating the difference in binding pose between cefadroxil and amoxicillin. **c** Initial steps in beta-lactam transport into the cell. Step 1 shows cefadroxil binding via the primary amine group to Glu622. Step 2 illustrates the movement of protons from His87 (TM2) to Glu56 (TM1), which releases Arg57 to clamp cefadroxil in the binding site. **d** Final step in drug release into the cell. Step 3 illustrates the protonation of Glu622, which weakens the interaction with the primary amine on the beta-lactam in the inward facing state, here modelled using AlphaFold (AF-Q63424-F1). Step 4 shows the deprotonation of Glu56, which results in Arg57 swinging back to engage the E53xxER motif. Step 5 is the release of the drug into the cytoplasm with two protons[95].

## Discussion

Solute carriers play essential roles in oral bioavailability and drug pharmacokinetics[67], and understanding how SLCs interact with drugs is essential for realizing the full potential of carrier-mediated drug delivery[68,69] and computer-aided drug design[9,70]. A key question concerning drug transport via PepT1 and PepT2 is how beta-lactam antibiotics are accommodated compared to physiological peptides. The recently reported structures of human PepT1 in complex with the dipeptide L-Ala-L-Phe (PDB:7PMX) in a similar extracellular open conformation enables a direct comparison to the cefadroxil and amoxicillin structures obtained in this study (Fig. 6a, b). Cefadroxil was the best substrate we tested, with an IC$_{50}$ of 19 µM compared to 32 µM for di-alanine, while amoxicillin was roughly ten times worse, at 270 µM (Fig. 1c). The difference in affinity between the two antibiotics can be rationalized from the structural comparison, with cefadroxil adopting a very similar pose to the di-peptide. Whilst the amino groups of all three molecules sit close to Glu622, it is only in cefadroxil where we observe overlay for the carbonyl and amide groups with the di-peptide. The carbonyl and methyl groups of the cepham ring also occupy the same position as the peptide carboxylate and phenylalanine side chain of the peptide, respectively. Of note is the interaction made between the cefadroxil carbonyl and Asn192, which helps to orientate the first side chain of peptides towards the binding site entrance, facilitating optimal interactions between the amino terminus, Asn348 and Glu622 to enable transport[33]. Although this orientation results in a small extension of the carboxylate in cefadroxil towards Arg57 and Lys161, the overall length of the drug is very similar to the di-peptide ligand of ~9 Å. The position of cefadroxil contrasts with the position adopted by amoxicillin, which, due to the stereochemistry of the sp3 hybridized carbon in the penem ring, is forced to extend the carbonyl group towards the extracellular gate, breaking the interaction with Asn192 and resulting in a suboptimal positioning of the drug in the binding

site. The position adopted by amoxicillin is very similar to that obtained for L-Phe-L-Ala from MD simulations, which also exhibits a lower affinity for PepT2 compared to L-Ala-L-Phe[33]. This enables amoxicillin to orientate the penem methyl groups towards Trp313 and Trp649 whilst maintaining the necessary interactions between the amino group and Glu622 and the carboxylate with Arg57 and Lys161. The hydroxyphenyl group of both antibiotics point into an acidic pocket dominated by Asp317 and sits close to the hydrophobic/aromatic pocket dominated by Trp313 and Trp649 that accommodates the side chain of the C-terminal amino acid in L-Ala-L-Phe. Similar pockets have been identified in both pro and eukaryotic POT family transporters and play important roles in dictating peptide specificity and affinity[71]. However, as noted above for the L-Phe-L-Ala peptide, utilizing the acidic pocket results in reduced affinity due to the distortions created in the peptide backbone, illustrated by amoxicillin. However, the stereochemistry of the cepham ring enables cefadroxil to utilize this pocket to accommodate the bulky R-group while maintaining optimal interactions observed in the L-Ala-L-Phe peptide structure. This provides a plausible explanation for why cefadroxil exhibits a lower IC$_{50}$ than the L-Ala-L-Phe peptide in our counterflow and Δµ$^{H+}$-driven uptake assays. Our analysis suggests that whilst PepT2 has evolved to recognize a diverse set of peptide substrates, this promiscuity comes at a cost to transport efficiency. Cefadroxil, however, can overcome these constraints and shows enhanced transport profiles compared to peptides, suggesting these pockets are a promising route for further prodrug development. Our results thus establish the core recognition and transport mechanism for beta-lactam antibiotics within the mammalian SLC15 family which can be exploited for future structure-based drug development programmes.

Finally, the results of the MD simulations demonstrate the close coupling between the orientation of the substrate with respect to Arg57 and the protonation of the E$^{53}$xxER glutamate residues,

particularly Glu56. Here, we observe that the cryo-EM poses of cefadroxil and amoxicillin (which show an interaction between Glu56 and Arg57, but not between Arg57 and the carboxyl group of the substrate) re-orient after protonation of Glu56. This takes the form of a salt-bridge swap: Glu56·H releases Arg57, which is then free to engage the substrate carboxyl group and clamp the ligand in the binding site ready for transport. Conversely, as discussed below, deprotonation of Glu56 would weaken the interaction with ligand by favouring the return of Arg57 to engage the E[53]xxER motif. We hypothesize that the functional role of ligand binding (Fig. 6c−step 1) is to promote the movement of protons from the extracellular gate at His87 further down into the transporter, which drives the closure of the extracellular gate (Fig. 6c−step 2)[33]. The cryo-EM poses likely represent trapped intermediates of this substrate-proton coupling mechanism. This hypothesis is further supported by our recent MD study into proton coupling within PepT2[64], where we found that an incoming peptide must engage Arg57 to disturb the pKa value of Glu56, thereby facilitating the movement of protons towards the E[53]xxER motif, as we observe for cefadroxil and amoxicillin (Fig. 4). This triggers the closure of the extracellular gate helices, TMs 1−2 and 7−8 and the movement of the transporter towards the inwards-facing orientation, before opening the intracellular gate, formed from TMs 4−5 and 10−11, where proton movement from Asp342 in the extracellular gate to Glu622 weakens the interaction with the amino terminus of the ligand (Fig. 6d −step 3)[64]. Deprotonation of Glu56 would similarly weaken the interaction with the carboxyl terminus (Fig. 6d−step 4), finally freeing the ligand to exit into the cytoplasm along with the two protons from Glu56 and Glu622 (Fig. 6d−step 5). These findings complement the results presented in this study, establishing a firm basis for the proposed salt-bridge swap mechanism of Arg57 engagement and delineating the role of the conserved E[53]xxER motif within the POT family. By further considering our comparison between substrate and inhibitor classes of beta-lactam antibiotics, we suggest that the engagement of Arg57 is one of the points in the transport cycle at which good and bad substrates are distinguished. In effect, our study suggests it is not the binding free energy that is predictive of drug transport but rather the ability of a drug molecule to participate in the dynamic transport mechanism originally optimized for physiological peptides.

## Methods

### Expression and purification

*Rattus norvegicus* SLC15A2 was expressed with a C-terminal his tagged GFP as previously described[33]. In brief, HEK293F cells were cultured in suspension in FreeStyle™ 293 Expression Medium (Life technologies Cat. no. 12338026) at 37 °C and 8% CO_2. Prior to transfection, with PEI-MAX (Polyscience Cat. No. 24765-1), cells were passed at a density of $7 \times 10^5$ cells/mL to give a density of $1.3–1.4 \times 10^6$ cells/mL for transfection. Sodium butyrate was added at 8 mM final concentration before transfection. Cells were returned to the incubator and harvested 36 hours post-transfection and frozen until required. Membranes were prepared by lysing the cells via sonication and unbroken cells and cell debris were pelleted at 10,000 g for 10 min at 4 °C and membranes were harvested through centrifugation at 200,000 g for one hour and washed once with 20 mM HEPES pH 7.5, 20 mM KCl. After washing the membranes were resuspended in PBS and snap frozen for storage at −80 until required.

For purification thawed membranes (~10 g wet weight) were solubilized in 130 mL of buffer containing 1% DDM: CHS (5:1 ratio) 1 × PBS, supplemented with an additional 150 mM NaCl and 10% glycerol containing for 90 min at 4 °C under gentle agitation using a magnetic stir plate. Insoluble material was removed through centrifugation for one hour at 200,000 g. PepT2 was purified to homogeneity using standard immobilized metal-affinity chromatography protocols in *n*-dodecyl-β-d-maltopyranoside (DDM) detergent (Anatrace Cat. No. D310LA) with cholesterol hemisuccinate (Merck Cat. No. C6512) (5:1 ratio DDM: CHS). In brief, 4 mL of nickel NTA resin (Fisher Scientific Cat. No. 10038124) was added with 25 mM imidazole for three hours at 4 °C under gentle agitation using a magnetic stir plate. The resin was loaded onto a gravity flow column (BioRad) and washed with eight column volumes (CVs) of buffer containing 0.15 % DDM: CHS (5:1 ratio) and 25 mM imidazole, followed by 15 CVs of buffer with 30 mM imidazole. The protein was eluted in four CVs of buffer containing 250 mM imidazole and dialysed overnight with TEV protease (1:0.5 M ratio) against 20 mM Tris pH 7.5, 150 mM NaCl with 0.02% DDM and 0.004% CHS at 4 °C under gentle agitation using a magnetic stir plate. Following TEV cleavage, the protease and cleaved his tagged GFP were removed through nickel affinity chromatography as described above adding 10 mM imidazole to the dialysed sample. The protein was then concentrated to 500 μL using a 50 KDa MWCO spin concentrator (Sartorius) at 4 °C and subjected to size exclusion chromatography (Superdex 200 increase, VWR Cat. No. 28-9909-44) in a buffer consisting of 20 mM Tris pH 7.5, 150 mM NaCl with 0.02 % DDM and 0.004% CHS.

### Reconstitution into liposomes

PepT2 was reconstituted into liposomes consisting of a 3:1 POPE:POPG (Avanti polar lipids, USA, Cat. No. 850757 C, 840457 C) using biobeads. The lipids were dried to obtain a thin film using a rotary evaporator and washed twice in pentane before being resuspended at 10 mg mL$^{-1}$ in lipid buffer (50 mM potassium phosphate at pH 7.5). These lipid vesicles were frozen and thawed twice in liquid nitrogen and stored at −80 °C until required. For reconstitution, the lipids were thawed and then extruded first through a 0.8-μm filter and then through a 0.4-μm filter. Purified PepT2 at 0.3 μg μl$^{-1}$ was added to the lipids at a final lipid:protein ratio of 100:1 (w/w) for transport assays or 10:1 for solid support membrane (SSM) experiments and incubated for 1 h at room temperature, then for a further 1 h on ice. After this time, biobeads were added in batches. After 24 h biobeads were removed and the proteoliposomes harvested by centrifugation at 120,000 g for 40 min before resuspension in lipid buffer at a final protein concentration of 0.25 μg μl$^{-1}$, or 0.5 ug ul$^{-1}$ for liposomes for SSM. They were subjected to three rounds of freeze−thawing in liquid nitrogen before storage at −80 °C.

### IC_50 calculations

Proteoliposomes were harvested through centrifugation before resuspending in inside buffer (120 mM potassium acetate, 2 mM MgSO_4 and 20 mM HEPES pH 7.5) and were subjected to four rounds of freeze thawing in liquid nitrogen to fully distribute the buffer and then extruded through a 0.2-μm filter. The proteoliposomes (equivalent of 1 μg protein per concentration) were diluted into external buffer (120 mM NaCl, 2 mM MgSO_4 and 20 mM HEPES pH 7.5) containing increasing concentrations of peptide or antibiotic and a trace amount of $^3$H di-alanine (Hartmann Analytic Cat. No. ART0893). The reaction was initiated through the addition of valinomycin at 1 μM and stopped after 4 minutes by rapidly filtering onto 0.22-μm filters, which were then washed with $2 \times 2$ mL cold water. The amount of peptide transported inside the liposomes was calculated by scintillation counting in Ultima Gold (Revvity, Cat. No. 6013326) with comparison to a standard curve for the substrate. Experiments were performed three times (from at least 2 independent purifications and reconstitutions) to generate an overall mean and s.d.

### Counterflow experiments

Proteoliposomes were harvested through centrifugation before resuspending in counterflow buffer (50 mM potassium phosphate pH 7.5) containing 0.5 mM peptide or antibiotic (unless stated otherwise) or water as the negative control and were subjected to four rounds of freeze thawing in liquid nitrogen to fully distribute the buffer and ligand. The proteoliposomes were then extruded through a 0.2-μm

filter. Transport was initiated by diluting into counterflow buffer containing 40 μM di-alanine with trace amounts of [3]H di-alanine, transport was allowed to proceed for 5 minutes before termination of the experiments through rapidly filtering onto 0.22-μm filters, which were then washed with $2 \times 2$ mL cold water. The amount of peptide transported inside the liposomes was calculated as above. Experiments were performed a minimal of five times and plotted as the level of transport for each ligand compared to the level observed with di-alanine as a percentage.

## SSM-based electrophysiology assays

SSM-based assays were performed on a SURFE$^2$R N1 (Nanion Technologies) with sensors prepared as follows[58]. 50 μl of 0.5 mM 1-octadecanethiol solution in isopropanol was added to the sensor and left overnight. The sensor was rinsed three times with isopropanol and then three times with water before being allowed to dry. 1.5 ul of 7.5 μg/μl 1,2-diphytanoyl-sn-glycero-3-phosphocholin in n-decane was applied to the surface of the sensor. Immediately after this 50 ul of non-activating buffer were added and left for an hour at room temperature. Proteoliposomes were diluted (5 ul liposomes with 36 ul buffer) in non-activating buffer (20 mM HEPES, 140 mM KCl, 2 mM MgCl$_2$ pH 7.2) and sonicated four times for 10 s in a water bath before application (2–6 μl) to the prepared 3 mm sensors. Sensors were incubated at room temperature and centrifugation at 2800 g for 30 min at 10 °C and incubated at room temperature for at least an hour prior to assaying. The activating buffer was made with the desired concentration of ligand (alanine, di-alanine, or the antibiotics of interest) in non-activating buffer. A single solution exchange workflow was used with activating buffer applied at 1 s and removed at 2 s. This technique was used to distinguish between antibiotics substrate (gives a current, similar to a known peptide control – di-alanine) and inhibitor (no current observed similar to a known non substrate, alanine) and raw traces from 1–3 s are shown in Fig. 1.

## Cryo-EM sample preparation and data acquisition

PepT2 post size-exclusion was mixed with 1 mM of the antibiotic for one hour prior to the addition of 1.2 molar excess of the nanobody (D8)[33] and incubated on ice for at least 30 min. The final concentration of PepT2 was between 5 and 6 mg mL$^{-1}$. The complex was adsorbed to glow-discharged holey carbon-coated grids (Quantifoil 300 mesh, Au R1.2/1.3) for 10 s. Grids were then blotted for 2 s at 100% humidity at 8 °C and frozen in liquid ethane using a Vitrobot Mark IV (Thermo Fisher Scientific).

Data were collected in counting mode in Electron Event Representation (EER) format on a CFEG-equipped Titan Krios G4 (Thermo Fisher Scientific) operating at 300 kV with a Selectris X imaging filter (Thermo Fisher Scientific) with slit width of 10 eV at 165,000× magnification on either a Falcon 4 or Falcon 4i direct detection camera (Thermo Fisher Scientific), with a physical pixel size of 0.693 Å (PepT2 + cefadroxil), 0.698 Å (PepT2 + amoxicillin) or 0.732 Å (PepT2 + cloxacillin). Movies were collected at a total dose of 54.8–57.6 e$^-$/Å2 fractionated to ~ 1 e$^-$/Å2 per frame.

## Cryo-EM data processing

Patched (20 × 20) motion correction, CTF parameter estimation, particle picking, extraction, and initial 2D classification were performed in SIMPLE 3.0[72]. All downstream processing was carried out in cryoSPARC 3.3.1[73] or RELION 3.1[74],using the csparc2star.py script within UCSF pyem[75] to convert between formats. Global resolution was estimated from gold-standard Fourier shell correlations (FSCs) using the 0.143 criterion and local resolution estimation was calculated within cryoSPARC.

The cryo-EM processing workflow for PepT2 with cefadroxil is outlined in Supplementary Fig. 1. Briefly, particles were subjected to two rounds of reference-free 2D classification ($k = 300$ each) using a

150 Å soft circular mask within cryoSPARC. Four volumes were generated from an 853,288 particle subset of the 2D-cleaned particles after multi-class ab initio reconstruction using a maximum resolution cutoff of 5 Å. These volumes were lowpass-filtered to 8 Å and used as references for a 4-class heterogeneous refinement against the full 2D-cleaned particle set (2,012,143 particles). Particles from the most populated and structured class were selected and non-uniformly refined against their corresponding volume lowpass-filtered to 15 Å, generating a 3.0 Å map. Bayesian polishing followed by per-particle defocus refinement and per-group CTF refinement (fitting beam tilt and trefoil) further improved map quality to 2.7 Å after non-uniform refinement. Alignment-free 3D classification using a soft spherical mask encompassing cefadroxil and surrounding partial TM helices was performed in RELION ($k = 4$, $T = 4$) resulting in one class with clear cefadroxil density. Particles belonging to this class were non-uniform refined against a 15 Å lowpass-filtered reference, generating a 3.0 Å volume with improved cefadroxil occupancy. An additional round of alignment-free 3D classification ($k = 3$, $T = 4$) followed by non-uniform refinement of the class with strongest cefadroxil density generated a 3.1 Å volume that was used for model refinements.

The cryo-EM processing workflow for PepT2 with amoxicillin is outlined in Supplementary Fig. 2. Briefly, particles were subjected to one round of reference-free 2D classification ($k = 300$) using a 150 Å soft circular mask within cryoSPARC. Four volumes were generated from a 618,157 particle subset of the 2D-cleaned particles after multi-class ab initio reconstruction using a maximum resolution cutoff of 7 Å. These volumes were lowpass-filtered to 8 Å and used as references for a 4-class heterogeneous refinement against the full 2D-cleaned particle set (2,022,970 particles). Particles from the two most populated and structured classes were selected and subjected to an additional round of multi-class ab initio to further purify the dataset. Particles from the two most prominent classes were combined (845,401 particles) and non-uniform refined against one of their corresponding volumes lowpass-filtered to 15 Å, generating a 3.2 Å map. Bayesian polishing followed by an additional round of 2D classification ($k = 200$) resulted in a selection of 582,083 pruned particles. These particles were non-uniform refined followed by CTF refinement (per-particle defocus refinement and per-group CTF refinement fitting beam tilt and trefoil) and another round of non-uniform refinement which generated a 2.9 Å volume. Alignment-free 3D classification using a soft spherical mask encompassing amoxicillin and surrounding TM helix side chains was performed in RELION ($k = 4$, $T = 20$) producing a class with strong ligand density. Particles belonging to this class were non-uniform refined against a 15 Å lowpass-filtered reference, generating a 3.0 Å volume with improved amoxicillin occupancy. An additional round of alignment-free 3D classification ($k = 4$, $T = 20$) followed by non-uniform refinement of the class with most resolved amoxicillin density generated a 3.2 Å volume that was used for model refinements.

The cryo-EM processing workflow for PepT2 with cloxacillin is outlined in Supplementary Fig. 5. Briefly, particles were subjected to one round of reference-free 2D classification ($k = 300$) using a 150 Å soft circular mask within cryoSPARC. Selected particles (1,795,057) were subjected to heterogeneous refinement against four 8 Å lowpass-filtered volumes generated ab initio from the amoxicillin dataset. Particles from the two most populated and structured classes were selected and subjected to a round of multi-class ab initio to further purify the dataset. Particles from the two most prominent classes were combined (756,906 particles) and non-uniform refined against one of their corresponding volumes lowpass-filtered to 15 Å, generating a 3.0 Å map. Bayesian polishing followed by per-particle defocus refinement and another round of non-uniform refinement generated a consensus 2.8 Å volume. Two alignment-free 3D classification schemes were performed in RELION ($k = 4$) using a soft spherical mask encompassing cloxacillin and partial TM helix side chains using a

regularization parameter ($T$) of either 20 or 10. For either T-value classification, particles were selected from the most populated classes, which also demonstrated the strongest ligand density. These particles were further subjected to non-uniform refinement against 15 Å lowpass-filtered references, followed by an additional round of alignment-free 3D classification. Particles were once again selected from the most populated and resolved classes and subjected to a final round of non-uniform refinement, generating volumes at resolutions of 3.1 Å and 2.9 Å.

### Model building and refinement

The model of rat PepT2 was docked into the globally-sharpened map for each drug complex and adjusted where necessary by manual building using Coot v. 0.9[76] and real-space refinement in PHENIX v. 1.20.1-4487[77] using secondary structure, rotamer and Ramachandran restraints. Ligand restraints were generated using Grade2[78]. The final models were validated using MolProbity[79] within PHENIX. Figures were prepared using UCSF ChimeraX v.1.7.1[80]. Figure 1a was created with BioRender.com.

### Unbiased molecular dynamics (MD) simulations

The protein models from cryo-EM were patched at the missing ECD loop using MODELLER[81], including residues 43-409 and 604-700 as a continuous chain in the processed model (as done by ref.[33]). We scored 200 models with QMEANDisCo[82] and selected the highest-scoring protein model for embedding into a 3:1 POPE:POPG bilayer of target size 10 × 10 nm (213/71 lipid molecules) with the CHARMM-GUI membrane builder[83]. Terminal groups were patched with ACE/NME residues in PyMOL, and the Glu53 / Glu56 residue protonation states were assigned as required in the GROMACS 2021[84] pbd2gmx tool. Note that whilst pKa predictors, such as PROPKA[85], suggests values of 8.76 and 5.16 for Glu53 and Glu56 respectively, it is also known that predictions that involve conformational change during the protonation state are much more inaccurate, and therefore we placed little credence on these values going forward. The protein and lipids were parametrized in the AMBER ff.14 sb[86] and slipids[87] forcefields respectively. The ligand parameters were obtained using the OpenFF 2.0 (SAGE) forcefield[88]. Gromacs was then used to solvate the system with 0.15 M NaCl and ~23,000 TIP3P water molecules. The box size after energy minimization was 9.9 × 9.9 × 11.3 nm. From these boxes, we ran six replicates of equilibration and production MD at temperature 310 K and pressure 1 bar. The thermostat was the v-rescale thermostat[89] (separate temperature coupling groups for membrane and solvent), the Berendsen barostat was used for equilibration and the Parrinello-Rahman barostat used for production[90]. The equilibration protocol was: 200 ps NVT, 1 ns NPT (C-alpha restraints and ligand heavy atoms), 20 ns NPT (C-alpha restraints only). The production simulations without restraints were then run for 1 µs.

### Absolute binding free energy (ABFE) simulations

We ran ABFE simulations[91] for cefadroxil, amoxicillin and cloxacillin (pose one only) binding in the Glu56 unprotonated and protonated protein conditions. We derived Boresch restraints[92] from the last 200 ns of each of the 6-replicate unbiased simulations and picked the frame closest to the restraint centre as ABFE starting frames, using MDRestraintsGenerator[93]. For cefadroxil and amoxicillin, we made 4 further replicate starting poses from the unbiased replicate with the highest affinity by running 4 times additional 200 ns-long equilibrations and deriving new sets of Boresch restraints from them (we report mean ± standard deviation among these 5 replicates). For cloxacillin, we report the (single) highest binding affinity of the unbiased Glu56 unprotonated and protonated replicates, respectively. Our lambda-protocol was to first add Boresch restraints (for the complex thermodynamic leg only, the ligand side was calculated using the analytic formula; through values 0, 0.01, 0.025, 0.05, 0.075, 0.1, 0.2, 0.3, 0.4,

0.5, 0.6, 0.8, 1.0), then annihilate Coulomb interactions (even 0.1 spacings) followed by van der waals interactions (even 0.05 spacings). This gives 44 windows for the complex and 31 windows for the ligand sides of the cycle. At each lambda window, ran energy minimization, 200 ps NVT equilibration (310 K, stochastic dynamics integrator), 1 ns NPT equilibration (310 K, stochastic dynamics integrator, 1 bar, berendsen), then 30 ns (complex windows)/100 ns (ligand windows) production (Parrinello–Rahman) with replica exchange. We analysed the simulations using alchemlyb (https://github.com/alchemistry/alchemlyb) with the mbar estimator[94].

### Reporting summary

Further information on research design is available in the Nature Portfolio Reporting Summary linked to this article.

## Data availability

Atomic coordinates for PepT2 have been deposited in the Protein Data Bank under accession codes 9BIR (cefadroxil); 9BIS (amoxicillin); 9BIT (cloxacillin – pose 1); and 9BIU (cloxacillin – pose 2). The cryo-EM maps have been deposited in the Electron Microscopy Data Bank (EMDB) under accession codes EMD-44599 (cefadroxil); EMD-44600 (amoxicillin); EMD-44601 (cloxacillin pose 1); and EMD-44602] (cloxacillin pose 2). The plasmids for expression of ratPepT2 and the nanobody have been deposited in Addgene under ID 167988 and 167989. Previously published PDB codes referred to in the work are 7NQK and 7PMX. The source data for the molecular dynamic's trajectories can be downloaded from the OSF. The source data underlying Figs. 1c, e, 5a and Table 1 are provided as a Source Data file. Source data are provided with this paper.

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

## Acknowledgements

This research was funded in whole or in part by the Wellcome Trust 215519/Z/19/Z and 219531/Z/19/Z. For the purpose of Open Access, the author has applied a CC BY public copyright licence to any Author Accepted Manuscript version arising from this submission. SML is a Wellcome Trust PhD student (218514). Computing was supported via the Advanced Research Computing facility, Oxford, the EPSRC ARCHER2 UK National Supercomputing Service and JADE (EP/X035603/1) granted via the High-End Computing Consortium for Biomolecular Simulation, (HECBioSim - http://www.hecbiosim.ac.uk), supported by EPSRC (EP/L000253/1). This research was funded (in part) by the Intramural Research Program of the NIH (to SML).

## Author contributions

J.L.P. & S.N. conceived the project. G.K. maintained cell stocks and undertook large-scale expression and tissue culture. J.L.P. performed all protein preparation, transport, and biochemical assays. J.C.D. and S.M.L performed all cryo-EM sample processing, data collection and image analysis. J.C.D., S.M.L., S.N. constructed the atomic models. S. M. L. and P.C.B. performed all molecular dynamics simulations and analysis. J.L.P., S.N. wrote the manuscript and prepared figures with contributions and discussions from S.M.L., P.C.B., J.C.D. and S.M.L.

## Competing interests

The authors declare no competing interests.
