## [Peer Review file · Nature Communications]

Structural basis for antibiotic transport and inhibition in PepT2.

Corresponding Author: Professor Simon Newstead

Version 0:

Reviewer comments:

Reviewer #1

(Remarks to the Author)

Parker et al report the Cryo-EM structures of *R. norvegicus* PepT2 in complex with three different β -lactam antibiotics, cefadroxil, amoxicillin and cloxacillin. This peptide transporter is responsible for the absorption of dietary di- and tripeptides in the kidney, though its characteristic promiscuity allows it to also recognize certain β -lactam antibiotics, thus making PepT2 a promising target to improve the pharmacokinetics of orally delivered antibiotics. The present work deepens greatly our understanding of this transporter, using functional assays to characterize the degree to which several known β -lactam binders are substrates as opposed to competitive inhibitors. This functional data is then used to interpret the binding modes of the aforementioned β -lactam-bound PepT2 structures. Crucially, Parker et al identify cefadroxil, the best substrate out of the tested β -lactams, as occupying a privileged pose in the binding vestibule that cannot be achieved by the poorer substrate amoxicillin due to the conformational constraints imposed by the 5-membered ring fused to the lactam. Furthermore, the researchers propose that the inhibitor cloxacillin owes its inhibitory activity to its lack of a primary amino group, which would prevent the proper positioning of the ligand between R57 (TM1) and E622 (TM10) to trigger transport. Further structural analysis of other β -lactam substrates and inhibitors reinforce this hypothesis. Finally, MD of antibiotic-bound PepT2 allow extrapolating the mechanism of peptide transport and proton coupling of PepT transporters to β -lactam antibiotics. The present manuscript is an exciting combination of structural biology, functional studies and computational simulations that provides great insight into the recognition mechanism of β -lactam antibiotics by PepT transporters, with potential implications for rational drug design to improve bioavailability. I propose the manuscript to be accepted with minor revisions.

- 1) In MD simulations cloxacillin does not adopt a stable binding pose, yet by Cryo-EM two distinct poses were able to be solved at good resolution. How can the MD and Cryo-EM data be reconciled? Do these two poses captured by Cryo-EM represent minimums of energy within the dynamic binding spectrum of this ligand?
- 2) Cefibuten is classified as an inhibitor in spite of having a primary amine group, and its lack of transport activity is attributed to the greater length of the peptide scaffold induced by the introduction of an aminothiazol group. Nevertheless, the primary amine is chemically quite distinct from that of a normal peptide or previously-studied β -lactam antibiotic substrates, being bound to a thiazol group which should lower its pKa considerably. Could the unique chemical properties of this 2-aminothiazol also play a part in preventing cefibuten from being a PepT2 substrate?
- 3) The authors might consider including in the discussion section a reference to the similarities between the cefadroxil and amoxicillin PEPT2 structures and the PEPT1 structure bound to the Ala-Phe dipeptide (ref. 34 in the manuscript). This could be done in the context of comparing these structures with the Ala-Phe-bound PEPT2 structure (ref. 33) (page 18; lines 202-5). I suggest this from the perspective of a non-expert in PEPTs, in order to highlight the robust structural and mechanistic similarities between PEPT2 and PEPT1. This would suggest that the pharmacophore information for β -lactams in PEPT2 reported in the submitted manuscript might likely be extended to PEPT1.
- 4) Figures 3 and 4 can be a bit difficult to follow due to the lack of labelling of TM numbers and the omission of labels for some of the key interactors mentioned in the manuscript. Thus, in figure 3 neither R57 nor E622 are labelled. Likewise, in figure 4 K161 in pose 1 and S626 in pose 2 are missing.
- 5) There is a typo error in the name AmXoxicillin in Fig. 2B.

Reviewer #2

(Remarks to the Author)

PepT2 is a major transporter for di- and tripeptides as well as peptidomimetic substrates, including certain widely used drugs, making it of considerable pharmaceutical interest. The manuscript by Parker et al. aims to establish the molecular basis for the transport and inhibition mechanism of β -lactam antibiotics by PepT2. Specifically, the authors have solved cryo-EM structures of the previously reported rat PepT2 in complex with three β -lactam antibiotics: amoxicillin, cefadroxil and cloxacillin. These structural findings are supported by functional assays and molecular dynamics (MD) simulations. The results presented are of interest for the scientific community and have the potential to enhance targeted drug design. Overall, the study is technically well executed, and the results are sound. The major advance and novelty here lie in the complex structures with different antibiotic compounds bound, providing molecular mechanisms for their varying effects on PepT2.

However, I have several specific demands and proposals, as well as open questions and comments, that need to be addressed to improve the clarity and impact of the work.

Major points

- The manuscript appears to be in a somewhat preliminary state and lacks elaboration, which has resulted in unnecessary additional work for the reviewer, e.g., it does not seem to have been properly proofread (see numerous points below). Please carefully consider these issues in case of a revision.
- The information in Figure 2 is relatively small for a single figure in the main manuscript. Please merge Figure 2 with Figure 3, as they also fits thematically.
- Figure 3 contains new, important information of this scientific contribution. However, the data and information are not provided optimally and only superficially. Please: i) provide two views for each antibiotic (currently binding pocket is only seen from one view point), ii) provide larger views - much too small currently, iii) indicate interactions by broken or dotted lines and, importantly, indicate distances in Angstrom (e.g., interactions at ≤ 3.5 Å for hydrophilic interactions - indicate such information in legend), iv) in all panels show density of ligand (in a brighter volume colour, now too dark): currently, the quality of this most important density cannot be estimated (see next comment), v) please also label helices, e.g., TM4, TM 8, TM10, etc. and vi) as the binding poses of both molecules are compared, it is necessary to display the views from same view angles.
- The main message of this manuscript is based on the information regarding the molecular binding mechanism of three antibiotics with the protein PepT2. Therefore and as already partially requested in the previous point, the quality of the ligand density is crucial to made reliable statements. Therefore and in addition to the previous request (Figure 3), the Reviewer kindly requests detailed and large views of the ligand densities with fitted corresponding antibiotic from at least two different angles (important) in a new Supplementary Figure. Focus should be put on the fitted small molecule (three different antibiotics and two different poses of cloxacillin) into the obtained ligand densities. This Suppl. Fig. should allow assessments of the quality of the density and reliability of the performed fit of the small molecules into the densities.
- Please provide a Table of the protein-ligand interactions of the three antibiotics (and two different poses of cloxacillin) as Supplementary information for direct comparison including also distances of interactions in Angstrom. This central information of the presented work needs to be communicated and displayed significantly better.
- Please consider the points mentioned above for Figure 3 (points i-vi), also for Figure 4.
- The sentence in lines 247-250 makes a strong statement. However, the provided experimental evidence is limited. Please provide more in vitro experimental evidence to keep this strong statement or tone down the statement.
- I suggest deleting the “, the mammalian proton-coupled peptide transporter” in the title and rather go for something like “Structural basis for antibiotic transport and inhibition in mammalian PepT2” to emphasize the findings of this study.

Further points and questions

- Please mention once in Abstract that the studied PepT2 protein is from rat.
- Sentence in lines 82-86: This sentence is about affinities and K_i values. However, the indicated Fig. 1B displays structures of antibiotics. This is misleading. If showing structure in this Figure, please bring Figure in this context (text main manuscript). Furthermore, the structure are relatively small and not optimally visible: Please make them larger and display in Supplementary Information, not in main manuscript.
- Figure 1C: Please indicate standard deviations or SEM and indicate accordingly in text. Furthermore, please also provide information about the intervals of confidence of the displayed four curves as well as statistics, which are currently not provided, e.g., $n=?$ and how many independent experiments per sigmoidal curve? From independent protein purifications and reconstitutions? Currently, information very scares.
- Figure 1: In general, statistics and corresponding information are currently not provided for panels C and D in Figure legend. Please provide information (e.g., triplicates, from how many independent experiments, etc.). Furthermore, introduce abbreviations used in the Figure (in legend), e.g., LZNP, etc. In addition, the concentration applied in panel D are not indicated, except for Cloxacillin (10 mM). In general the legend to this Figure is too minimalistic: please provide pertinent information for the reader.
- Please consider the previous two points mentioned for Figure 1 also for Figure 6 (e.g., missing statistics and information). Furthermore, also move these structures to Supplementary Information (as for Figure 1B).
- Please carefully read manuscript: there are for a manuscript submitted to a journal noticeably many typos and flaws. Only to mention a few, see Figure 2: "Extraracellular Gate" instead of "Extracellular Gate", "Amoxicillin" instead of Amoxicillin" and the colour scale is missing (bottom, right: electrostatic potential scale as well as no information is provided on the indicated numbers, e.g., is this kT/e ?
- Please move Table 1 to Supplementary Information: This is technical information.
- Figure 3: i) R57 and E622 are discussed, but not shown in Fig. 3A. Please clearly indicate interactions of R57 with the carboxylate and E622 with the primary amine group (consider my comment above of showing two different view, this will make display of such, currently missing, interactions possible).

- In Figures the one letter and in the text the three letter codes (or full names) of amino acids are used. This is suboptimal, please use one code only (probably one letter code is best, in this given situation).
 - Sentence in lines 188-190: Why is L650 not mentioned? Considering Fig. 3D, this is confusing.
 - Please provide estimated pKa values for the in "Interplay of protonation and ligand recognition" mentioned amino acid residues, e.g., E53, E56, etc. and provide context for the obtained numbers, e.g., do the obtained values fit within the proposed context provided by the in vitro experiments (performed at a given pH) and the conducted MD experiments? For pKa value estimation, several freely accessible servers are available.
 - Figure 7: Panels A and B are confusing and short in information. Regarding the former, the overlay of L-Ala-L-Phe, Amoxicillin and Cefadroxil is confusing: protein ligand interaction are not optimally discernible. Please improve this, e.g., by showing the ligands at two different views separately and next to each other. Furthermore, add broken or dotted lines for interaction and distances in Angstrom (see my comments above regarding Figure 3).
 - Sentence in lines 128-130: I would appreciate a comment on the action of the used nanobody, especially its inhibitory behaviour. According to the previous publication (PMID: 34433568), nanobody (D8) is "capable of inhibiting transport". A reader unaware of this study might be confused (inhibitory nanobody and inhibitory antibiotic) and imply a potential conceptual issue.
 - Is there a rational why the antibiotic ligands were added to the cryo-EM samples at rather low concentrations (i.e., 1 mM final concentration)? Considering the IC50 values of 19 μM and 270 μM for cefadroxil and amoxicillin (measured in proteoliposomes), this corresponds only in a nearly fourfold molar excess for the detergent solubilized proteins, explaining the poor occupation and necessitating (potentially preventable) extensive alignment-free 3D classification.
 - Is there a reason for the low resolution of the extracellular domain (ECD)? It appears as if the same construct and the same nanobody were used as in the previous study (PMID: 34433568), where the ECD was clearly resolved (nanobody-mediated fixation of the ECD). Additionally, in most of the provided 2D classes density for the ECD is clearly visible (admittedly very cloudy, supporting its flexible nature). Line 135 states that the ECD "is much lower resolution", however, its not visible at all; therefore, you might want to alter this statement.
 - Line 139: The 562 C-alpha atoms suggest that rather the whole transmembrane domains were used for alignment, i.e., including the loops connecting the TM helices. Please check.
 - Figure 4 legend for panel (A) reads wrong ("showing the three orientations overlaid in the binding site") – only two poses are overlaid.
 - Based on the density in Fig. 4B and C, the chlorophenyl group would also fit when rotated by 180° around the bond to the methylisoxazole moiety: Is there any evidence for the current placement?
 - What is the r.m.s.d. with the peptide-bound structure of human PepT2 (PDB:7PMX) structure? Please indicate.
 - Sentence in lines 373-375 "...is consistent with the three binding poses..." is confusing: I thought there only two poses observe in cryo-EM structure?
 - Line 406: not consistent with chapter one stating "cefadroxil displayed an IC50 of 19 μM ".
 - Generally, the purification procedure (lines 501 - 509) is not explained sufficiently: Please provide more details.
 - Line 577: Please verify if the following statement: " $\sim 1 \text{ e}^- / \text{\AA}^2$ per frame" is correct.
 - The carboxyl group in the schematic compound representation for cefadroxil and amoxicillin in Fig. 3B and D are deprotonated (as expected at pH 7.5), however, in the corresponding Fig. S5 for cloxacillin the carboxyl group are in a protonated state. Please improve.
 - Supplementary Information Figs. 1, 2 and 4: Certain 3D classifications yield distributions percentages above 100%, potentially due to rounding errors (e.g., focussed 3D classification in Fig. S1 yields 100.1%) – please correct.
- Minor point: Spelling issues / wording
- Line 75: "binding site" not "biding site"
 - Line 77: The reviewer understands "inhibition mechanism", but what is meant by "substrate", respectively, "substrate mechanism" ? Probably, the authors mean "substrate binding and inhibition mechanisms"?
 - Sentence in lines 82-86: Quite long sentence and doesn't read nicely, please improve.
 - Line 41: delete anaerobic bacteria
 - Please make the "i" in the inhibitory constant lowercase, i.e., Ki
 - Figure 1c, bar graph: the unit is μM instead of uM
 - Line 104: Should read "SURF2R"
 - Line 187: remove the space in $\sim 90^\circ$
 - Line 225: "...E53xxER..." - unclear what the number 53 stands for. Please provide information to reader.
 - Line 543: 3H - 3 is currently not superscript
 - Line 546: full stop
 - Line 516: delete DDM:CHS
 - Line 501: Unclear - was 1 x PBS supplemented with 150 mM NaCl or does your PBS include 150 mM NaCl?
 - Methods: For consistency, either indicate centrifugation speed as "x g" or simply "g"
 - Line 505: reads like a repetition from above.
 - Line 554: " μl " instead of "ul"
 - Line 372, 372 and 581: spaces are missing
 - Line 418: "sp3 hybridized"
 - Line 620: should read "non-uniformly refined"
 - Line 511: indicate if molar ratio or w/w for the lipids
 - Line 553: Hepes should be capitalized for consistency

Reviewer #3

(Remarks to the Author)

The manuscript titled "Structural basis for antibiotic transport and inhibition in PepT2, the mammalian proton-coupled peptide transporter." by Joanne L. Parker, et al combines protein-ligand structural analysis with cryo-em and molecular dynamics (MD) simulations to understand the antibiotics transport and inhibition mechanism in PepT2 transporter. Initially PepT2 is reconstituted into liposomes to compare three different antibiotics in their effect on alanine dipeptide uptake through IC50 experiments. These studies were complemented with counterflow experiments. These showed that cefadroxil and amoxicillin act as ligands while cloxacillin acts as a competitive inhibitor. They solved cryo-em

structures of PepT2 bound with three beta-lactam antibiotics - cefadroxil, amoxicillin and cloxacillin, discovered transport mechanism of the former two ligands and provide rationale for cloxacillin functioned as inhibitor instead. When solving the structure bound to cloxacillin, once more the protein structure is close to Apo PepT2. However, different poses are identified for the ligand: one of them establishes interactions similar to the previous ligands, but the possibility of the second binding mode can possibly result in inhibitory behavior. It would be interesting to see based on recent CryoEM methods (e.g Pilar Cosio, Gerhard Hummer) what the population of each pose is.

They performed unbiased MD simulations and absolute free energy calculations for both deprotonated and protonated systems. The computations complement their conclusions from experiments for discovering the role of key interactions in transporting cefadroxil and amoxicillin substrates, and help support their hypothesis of protonation mechanism in antibiotic binding and transport. Further analysis of additional substrates and inhibitors provides structural insights in distinguishing substrates and inhibitors.

Standard MD simulations complement the CryoEM by using the CryoEM starting poses and different protonation states. I wonder if constant pH simulations would have been better, since they would allow the protonation/deprotonation of GLU53 and 56 in response to the instantaneous environment while keeping the pH at 7.5. For example, Jana Shen has recently shown that more than the state (protonated/deprotonated) it is the ability to deprotonate which can differ from ligand to ligand.

Page 15 Table 2, (1) cloxacillin has no error bar compared to the other two, the authors mentioned in the Methods part that only the highest binding affinity replicate is reported for cloxacillin, they should include the rationale behind. They include MBar estimation of all replicates in SI since the mbar estimation for each replicate should have error bar, which are good for checking the consistency of replicates. (2) the authors should provide appropriate reasons for the difference in rankings of binding affinity among cefadroxil, amoxicillin, cloxacillin (in IC50 experiments: cefadroxil > cloxacillin > amoxicillin, while in computations and previously reported Ki values: cefadroxil > amoxicillin > cloxacillin).

Page 21 line 486-487, the authors suggest that binding affinity is not predictive for drug transport. This conclusion is not consistent throughout the manuscript. In page 3 line 96, they suggest higher transport efficiency is consistent with higher IC50. From the section starting from page 16 line 340, both high efficient substrates - cefadroxil and cefaclor, have high IC50, and inhibitors have low IC50 in general except for the low transport efficiency substrate ampicillin. Apparently the datapoints are not enough to simply draw that conclusion.

Minor:

Figure 2B there is a typo in "amxoxicillin"

Page 3 line 75, typo, "biding" should be binding.

Page 6 Figure 2, needs explanations for the scale bar "-12.46-0-16.22" in caption.

Page 10 Figure 4 caption, "...showing the three orientations..." should be "...two orientations...".

Page 13 Figure 5C, need explanations of red/blue colors in caption.

Page 15 line 332-333, the authors suggest that there is no stabilization of binding upon Glu56 protonation, they should include figure/data to support the conclusion.

Page 17 line 375, "...three binding poses..." should be "...two binding poses...".

Page 19 Figure 7D, the authors should indicate the source of "Inward Facing" structural model.

Reviewer #4

(Remarks to the Author)

I co-reviewed this manuscript with one of the reviewers who provided the listed reports. This is part of the Nature Communications initiative to facilitate training in peer review and to provide appropriate recognition for Early Career

Researchers who co-review manuscripts.

Version 1:

Reviewer comments:

Reviewer #1

(Remarks to the Author)

I am satisfied with the changes introduced in the new version of the manuscript. I propose to accept the manuscript in the present form for publication in Nat Commun.

Reviewer #2

(Remarks to the Author)

We appreciate the author's efforts to improve the manuscript. While most revisions were properly done in accordance with the Reviewers' requests, the manuscript still requires careful revisions.

- The Reviewer requests a thorough proofreading; there are still several spelling errors and inconsistencies (e.g., IC50 and IC50 (50 subscript, mL and ml, etc.) throughout the manuscript that were not fully corrected. Here a few examples (this list is not exhaustive):

- Line 101 (and elsewhere): The "2" in Lys[Z(NO₂)]-Pro should be lowercase
 - Figure 1c, bar graph: the unit is μ M instead of uM
 - Line 136: A space is missing in "TMs10-11"
 - Line 138: "coulomb" should be capitalized
 - Line 198: A letter is missing in "Ly161"
 - Lines 383-385: inconsistencies with spaces between numbers and units
 - Line 436: Mixture of one and three-letter codes of amino acids in the same sentence, i.e., the one-letter code was used for K161 instead of three-letter code (please stay consistent)
 - Line 538: "50KDa" should read 50 kDa
- ...

- Figure 1c: Unfortunately, not all of my points were addressed in this revision. Please provide information about the confidence intervals for the four representative curves displayed (left panel). Additionally, please include the standard deviations (SD) for the individual data points shown (left panel; the right panel is OK). This will help the reader understand the variation of the individual points at different concentrations. Furthermore and based on the added "n=3" statement in the revised version: Are these technical replicates of the representative IC50 curves? Finally, why not showing a merge of at least three biological replicates here, instead of "representative IC50 curves"? Data seem to be available.

- Figure 5b: Please consider the points in my previous comment.

- Considering the ligand densities: I agree that the correct modelling of ligands in cryo-EM maps at the achieved resolution is tricky and I appreciate the authors statement that the modelled pose is "is just one of several possible positions". Especially in the case of pose 2 for the cloxacillin-bound structure (Supplementary Figure 6d), considerable portions of the compound are not within the density at the shown threshold. Considering these modelling uncertainties, lines 233-238 should highlight and contain your statement "is just one of several possible positions".

- The arrows used in multiple Figures as hydrogen bond representatives are rather confusing (might indicate donors or acceptors, unclear) and distracting: Please replace with dotted or dashed lines.

- Fig. 2b and c (and also in Suppl. Information): Visualization and measurement of H-bonds is inconsistent. Normally, H-bond distances are indicated and measured between heavy atoms (e.g., Y61 with carbonyl oxygen atom of cefadroxil). However, in the same panel (b in Fig. 2), distances are measured from heavy atoms to hydrogen atoms, e.g., D317 and E622 to ligand H-atoms (instead to ligand heavy atoms, i.e., nitrogen and oxygen atoms). Please carefully adapt concerned Figures and, importantly, check measured distances: It might be needed to perform major changes in distances including in other Figures and in the text, e.g., the D317 to OH-group in Fig. 2d is with 2.12 Å probably too short and not measured in the usual convention.

- Figure 1: The colouring of the compounds is suboptimal – sulphur and nitrogen atoms are barely visible, and the compounds are rather small. Please revise and make use of the empty space within this figure.

- Figure 3c: Please add the text "EM density" (as in b) for consistency. Furthermore, please add the membrane boundaries in panel a as illustrated in Fig. 2a and mark the cytoplasm and extracellular sides (also as in Fig. 2a).

- The newly added encircled TM labels (combined with the filled ellipses and amino acid residue labels) overload the respective panels and distract from the actual topic, i.e., the protein-ligand interactions. Please tone them down, e.g., by reducing the sizes of the filled circles and ellipses.

- Figure 5a: Concentrations for all compounds should be added (i.e., also for Ala-Ala and Cefaclor). Furthermore, "BenzylPenicillin" spells Benzylpenicillin. As a side note: The data points for Ala-Ala are coloured inconsistently.
- Supplementary Figure 6a, left panel: The interaction between Y61 and the sulphur atom is not shown and K161 is not labelled. Please also check panel b for inconsistencies and missing information.
- Supplementary Figure 7 and 8: Please colour plots darker for enhanced contrast and better visualization.
- Figure 2d: A typical distance between donor and acceptor in salt bridges is about 2.8 Å. The distances between K161 and R57 from the carboxylate of the ligand are both > 4 Å. Therefore, I ask myself if such interactions are significant and no interactions should be drawn in this panel/Figure. Furthermore, I suggest to indicate only decimals, e.g., 2.5 Å instead of 2.52 Å.
- Figure 6c: In the legend it's called "E53xxER", in the panel only "ExxER". Besides of being inconsistent, please provide information about the meaning of "53" to the reader.
- Figure 1 legend for panel e: For completeness, please state that the non-transported alanine and cloxacillin were tested as well, as the figure displays SSM traces for these compounds.
- Please capitalize "Figure" in all supplementary figure legends.
- Line 605: "... 10 e-V ..." - the superscripted minus sign should be removed (the unit for electron-volt is eV)

Reviewer #3

(Remarks to the Author)

The authors have successfully addressed my initial comments.

Reviewer #4

(Remarks to the Author)

REVIEWER COMMENTS

We would first like to thank the referees for taking the time to read and comment on our manuscript and acknowledge the considerable extra time and effort this work takes. We are confident the comments and suggestions improved the study.

Reviewer #1 (Remarks to the Author):

Parker et al report the Cryo-EM structures of *R. norvegicus* PepT2 in complex with three different β -lactam antibiotics, cefadroxil, amoxicillin and cloxacillin. This peptide transporter is responsible for the absorption of dietary di- and tripeptides in the kidney, though its characteristic promiscuity allows it to also recognize certain β -lactam antibiotics, thus making PepT2 a promising target to improve the pharmacokinetics of orally delivered antibiotics. The present work deepens greatly our understanding of this transporter, using functional assays to characterize the degree to which several known β -lactam binders are substrates as opposed to competitive inhibitors. This functional data is then used to interpret the binding modes of the aforementioned β -lactam-bound PepT2 structures. Crucially, Parker et al identify cefadroxil, the best substrate out of the tested β -lactams, as occupying a privileged pose in the binding vestibule that cannot be achieved by the poorer substrate amoxicillin due to the conformational constraints imposed by the 5-membered ring fused to the lactam. Furthermore, the researchers propose that the inhibitor cloxacillin owes its inhibitory activity to its lack of a primary amino group, which would prevent the proper positioning of the ligand between R57 (TM1) and E622 (TM10) to trigger transport. Further structural analysis of other β -lactam substrates and inhibitors reinforce this hypothesis. Finally, MD of antibiotic-bound PepT2 allow extrapolating the mechanism of peptide transport and proton coupling of PepT transporters to β -lactam antibiotics. The present manuscript is an exciting combination of structural biology, functional studies and computational simulations that provides great insight into the recognition mechanism of β -lactam antibiotics by PepT transporters, with potential implications for rational drug design to improve bioavailability.

I propose the manuscript to be accepted with minor revisions.

1) In MD simulations cloxacillin does not adopt a stable binding pose, yet by Cryo-EM two distinct poses were able to be solved at good resolution. How can the MD and Cryo-EM data be reconciled? Do these two poses captured by Cryo-EM represent minimums of energy within the dynamic binding spectrum of this ligand?

The reviewer raises an interesting point and one that sometimes comes up when comparing structural work directly with MD. There are several possibilities that might explain the discrepancy. For example, it is known that membrane proteins are very sensitive to the nature of the lipid environment (see for example – Dalal et al. 2024 Nat Comms 15:25) and thus over-stabilization of certain poses could occur. Conversely, the force-fields for small molecules are far from perfect and for weakly binding entities, such as the beta-lactam drugs in this study, any subtle difference will be acutely manifested. Given these points, we have altered the manuscript to say that the results are qualitatively consistent, in that less stability in the MD correlates with a less clearly defined single cryo-EM pose.

2) Ceftibuten is classified as an inhibitor in spite of having a primary amine group, and its lack of transport activity is attributed to the greater length of the peptide scaffold induced by the introduction of an aminothiazol group. Nevertheless, the primary amine is chemically quite distinct from that of a normal peptide or previously-studied β -lactam antibiotic substrates, being bound to a thiazol group which should lower its pKa considerably. Could the unique chemical properties of this 2-aminothiazol also play a part in preventing ceftibuten from being a PepT2 substrate?

It may well be possible that the unique properties of a 2-aminothiazol could lower its pKa and thus hinder its properties in terms of being a substrate. However, the most likely explanation is simply that the amine group is not optimally positioned to make the necessary contact with E622. Indeed, one might expect that even as the neutral amine form, this would still interact favourably with E622. To test that, one would need analogues that use the aminothiazole in way that positions the amine group in the equivalent position as found cefadroxil/amoxicillin. We are not aware of analogues that do this currently, but it could be worth exploring in the future.

3) The authors might consider including in the discussion section a reference to the similarities between the cefadroxil and amoxicillin PEPT2 structures and the PEPT1 structure bound to the Ala-Phe dipeptide (ref. 34 in the

manuscript). This could be done in the context of comparing these structures with the Ala-Phe-bound PEPT2 structure (ref. 33) (page 18; lines 202-5). I suggest this from the perspective of a non-expert in PEPTs, in order to highlight the robust structural and mechanistic similarities between PEPT2 and PEPT1. This would suggest that the pharmacophore information for β -lactams in PEPT2 reported in the submitted manuscript might likely be extended to PEPT1.

We agree with the reviewer and indeed, the PepT1 structure is in a similar outward open state to our PepT2 structure. We have now amended the comparison figure to include the structure of the L-Ala-L-Phe dipeptide in PepT1 (PDB:7PMX) and noted this in the text. Incidentally, the r.m.s.d between the rat PepT2 structure reported in our study and the human PepT1 structure reported in Killer et al., 2021 is 1.18 Angstroms over 377 C α atoms, which represents the transmembrane domain.

4) Figures 3 and 4 can be a bit difficult to follow due to the lack of labelling of TM numbers and the omission of labels for some of the key interactors mentioned in the manuscript. Thus, in figure 3 neither R57 nor E622 are labelled. Likewise, in figure 4 K161 in pose 1 and S626 in pose 2 are missing.

We have now added TM labels for these figures.

5) There is a typo error in the name AmXoxicillin in Fig. 2B. – corrected.

Reviewer #2 (Remarks to the Author):

PepT2 is a major transporter for di- and tripeptides as well as peptidomimetic substrates, including certain widely used drugs, making it of considerable pharmaceutical interest. The manuscript by Parker et al. aims to establish the molecular basis for the transport and inhibition mechanism of β -lactam antibiotics by PepT2. Specifically, the authors have solved cryo-EM structures of the previously reported rat PepT2 in complex with three β -lactam antibiotics: amoxicillin, cefadroxil and cloxacillin. These structural findings are supported by functional assays and molecular dynamics (MD) simulations.

The results presented are of interest for the scientific community and have the potential to enhance targeted drug design. Overall, the study is technically well executed, and the results are sound. The major advance and novelty here lie in the complex structures with different antibiotic compounds bound, providing molecular mechanisms for their varying effects on PepT2.

However, I have several specific demands and proposals, as well as open questions and comments, that need to be addressed to improve the clarity and impact of the work.

Major points

- The manuscript appears to be in a somewhat preliminary state and lacks elaboration, which has resulted in unnecessary additional work for the reviewer, e.g., it does not seem to have been properly proofread (see numerous points below). Please carefully consider these issues in case of a revision.

- The information in Figure 2 is relatively small for a single figure in the main manuscript. Please merge Figure 2 with Figure 3, as they also fits thematically.

We agree with the referee and our revised manuscript now includes a merged figure – new Figure 2.

- Figure 3 contains new, important information of this scientific contribution. However, the data and information are not provided optimally and only superficially. Please: i) provide two views for each antibiotic (currently binding pocket is only seen from one view point), ii) provide larger views - much too small currently, iii) indicate interactions by broken or dotted lines and, importantly, indicate distances in Angstrom (e.g., interactions at ≤ 3.5 Å for hydrophilic interactions - indicate such information in legend), iv) in all panels show density of ligand (in a brighter volume colour, now too dark): currently, the quality of this most important density cannot be estimated (see next comment), v) please also label helices, e.g., TM4, TM 8, TM10, etc. and vi) as the binding poses of both molecules are compared, it is necessary to display the views from same view angles.

We have modified the old Figure 3 to include labels for the helices and distances for the interactions between the drugs and the transporter binding site. We have also included different views of the drug binding site and ligands, with their corresponding cryo-EM density in a new Supplementary Figure S4.

- The main message of this manuscript is based on the information regarding the molecular binding mechanism of three antibiotics with the protein PepT2. Therefore, and as already partially requested in the previous point, the quality of the ligand density is crucial to make reliable statements. Therefore, and in addition to the previous request (Figure 3), the Reviewer kindly requests detailed and large views of the ligand densities with fitted corresponding antibiotic from at least two different angles (important) in a new Supplementary Figure. Focus should be put on the fitted small molecule (three different antibiotics and two different poses of cloxacillin) into the obtained ligand densities. This Suppl. Fig. should allow assessments of the quality of the density and reliability of the performed fit of the small molecules into the densities.

We agree and include different views of the density for all the drugs in the supplementary material to support the single views in the main text.

- Please provide a Table of the protein-ligand interactions of the three antibiotics (and two different poses of cloxacillin) as Supplementary information for direct comparison including also distances of interactions in Angstrom. This central information of the presented work needs to be communicated and displayed significantly better.

We accept the view of the referee on this point but feel the addition of the distances in the main text figures and supplementary figures address this point.

- Please consider the points mentioned above for Figure 3 (points i-vi), also for Figure 4.

We have modified the figures accordingly. Thank you.

- The sentence in lines 247-250 makes a strong statement. However, the provided experimental evidence is limited. Please provide more in vitro experimental evidence to keep this strong statement or tone down the statement.

Many thanks. We have changed 'reveals' to 'suggests'.

- I suggest deleting the "the mammalian proton-coupled peptide transporter" in the title and rather go for something like "Structural basis for antibiotic transport and inhibition in mammalian PepT2" to emphasize the findings of this study.

Agreed.

Further points and questions

- Please mention once in Abstract that the studied PepT2 protein is from rat. done

- Sentence in lines 82-86: This sentence is about affinities and K_i values. However, the indicated Fig. 1B displays structures of antibiotics. This is misleading. If showing structure in this Figure, please bring Figure in this context (text main manuscript). Furthermore, the structure are relatively small and not optimally visible: Please make them larger and display in Supplementary Information, not in main manuscript.

Agreed – we have moved the call out for Fig. 1b to avoid confusion. However, we do feel that the chemical structures for the antibiotics we are looking at are relevant and helpful to readers given the description of their differing affinities to PepT1 and PepT2.

- Figure 1C: Please indicate standard deviations or SEM and indicate accordingly in text. Furthermore, please also provide information about the intervals of confidence of the displayed four curves as well as statistics, which are currently not provided, e.g., $n=?$ and how many independent experiments per sigmoidal curve? From independent protein purifications and reconstitutions? Currently, information very scarce.

We had previously only mentioned the number of replicates in the methods and thank the reviewer for highlighting this point. We have now included the number of replicates in the figure legend (n=3). We have also added the number of purifications and reconstitutions to the methods.

- Figure 1: In general, statistics and corresponding information are currently not provided for panels C and D in Figure legend. Please provide information (e.g., triplicates, from how many independent experiments, etc.).

We previously included this information in the methods. As above, we thank the reviewer for noticing this absence and we have moved this to also be in the figure legend.

Furthermore, introduce abbreviations used in the Figure (in legend), e.g., LZNP, etc. In addition, the concentration applied in panel D are not indicated, except for Cloxacillin (10 mM). In general the legend to this Figure is too minimalistic: please provide pertinent information for the reader.

For clarity this information was provided in the methods but has now also been included in the legend and the abbreviation of LZNP has been written in full.

- Please consider the previous two points mentioned for Figure 1 also for Figure 6 (e.g., missing statistics and information). Furthermore, also move these structures to Supplementary Information (as for Figure 1B).

The number of replicates and compound concentrations used has now been added to the legend and detailed in the methods as appropriate. As noted above in our response to the chemical compounds in Fig. 1, we feel the inclusion of the chemical structures in the main text figure is helpful for readers to understand the key similarities and differences within the beta-lactam antibiotics studied in our paper.

- Please carefully read manuscript: there are for a manuscript submitted to a journal noticeably many typos and flaws. Only to mention a few, see Figure 2: "Extraracellular Gate" instead of "Extracellular Gate", "Amoxicillin" instead of "Amoxicillin" and the colour scale is missing (bottom, right: electrostatic potential scale as well as no information is provided on the indicated numbers, e.g., is this kT/e ?

done

- Please move Table 1 to Supplementary Information: This is technical information.

done

- Figure 3: i) R57 and E622 are discussed, but not shown in Fig. 3A. Please clearly indicate interactions of R57 with the carboxylate and E622 with the primary amine group (consider my comment above of showing two different view, this will make display of such, currently missing, interactions possible).

done

- In Figures the one letter and in the text the three letter codes (or full names) of amino acids are used. This is suboptimal, please use one code only (probably one letter code is best, in this given situation).

We can see the perspective of the reviewer on this point. However, we feel that when reading the main text, the three letter abbreviations make the text easier to read, whilst in figures they take up too much space. We would therefore like to keep the format we have chosen.

- Sentence in lines 188-190: Why is L650 not mentioned? Considering Fig. 3D, this is confusing.

Leucine 650 is part of the hydrophobic pocket and we do mention the side chain in this context.

- Please provide estimated pKa values for the in "Interplay of protonation and ligand recognition" mentioned amino acid residues, e.g., E53, E56, etc. and provide context for the obtained numbers, e.g., do the obtained values fit within the proposed context provided by the in vitro experiments (performed at a given pH) and the conducted MD experiments? For pKa value estimation, several freely accessible servers are available.

Indeed, several servers do exist for this, the most popular is PROPKA. However, we should state that it would be unwise to put too much weight against any such prediction against a single conformation. Furthermore, the performance of empirical pKa prediction tools is known to be optimal when no structural change occurs following the change in protonation (see discussion in <https://pubs.acs.org/doi/10.1021/acs.jctc.1c01257>). That is most definitely not the case here (and indeed is the very essence of the mechanism). In the structure when E56 is close to R57 then the pKa prediction for E56 will be very low, as one would expect. However, the fact that things can rearrange upon protonation means that protonation will be more favourable than it appears from the prediction. In our previous work (<https://elifesciences.org/reviewed-preprints/96507>) we were able to address some of this complexity - in that work with a peptide bound, we were able to see the shift in pKa via constant pH MD and orthogonal ABFE calculations. Whilst in this work we were not able to converge the constant pH simulations (a well known problem), we do observe that the ABFE values for cefadroxil have the same direction of shift (and indeed larger than with the peptide), so there is good reason to believe that the underlying thermodynamics are similar. Thus, in the manuscript, we mention in brief the pKa predictions in the Methods and the limitations and refer to this in the “Interplay of protonation and ligand recognition” section.

- Figure 7: Panels A and B are confusing and short in information. Regarding the former, the overlay of L-Ala-L-Phe, Amoxicillin and Cefadroxil is confusing: protein ligand interaction are not optimally discernible. Please improve this, e.g., by showing the ligands at two different views separately and next to each other. Furthermore, add broken or dotted lines for interaction and distances in Angstrom (see my comments above regarding Figure 3).

Although we appreciate the referee’s comment, we respectively disagree on this point. The main take home point concerning this figure is that when overlaid the drug molecules that are transported sit in the same position as the previously reported L-Ala-L-Phe dipeptide. In addition, as can be seen in panel B, the backbone of cefadroxil aligns more closely with the dipeptide compared with amoxicillin, which is likely to contribute to the more favourable IC50 values and ABFE values reported in our study. Panel A is to assist readers familiar with previous peptide transporter research from our and other groups in locating the drugs in the context of previously identified specificity pockets.

- Sentence in lines 128-130: I would appreciate a comment on the action of the used nanobody, especially its inhibitory behaviour. According to the previous publication (PMID: 34433568), nanobody (D8) is “capable of inhibiting transport”. A reader unaware of this study might be confused (inhibitory nanobody and inhibitory antibiotic) and imply a potential conceptual issue.

We have removed the word ‘inhibitory’ from the sentence to avoid confusion. Indeed, the inhibitory properties of the nanobody are not relevant to this study.

- Is there a rational why the antibiotic ligands were added to the cryo-EM samples at rather low concentrations (i.e., 1 mM final concentration)? Considering the IC50 values of 19 μ M and 270 μ M for cefadroxil and amoxicillin (measured in proteoliposomes), this corresponds only in a nearly fourfold molar excess for the detergent solubilized proteins, explaining the poor occupation and necessitating (potentially preventable) extensive alignment-free 3D classification.

Amoxicillin shows relatively poor solubility (20 mM at neutral pH) therefore this substrate could only be added in the low mM range. We therefore added amoxicillin and cefadroxil at the same concentration for consistency.

- Is there a reason for the low resolution of the extracellular domain (ECD)? It appears as if the same construct and the same nanobody were used as in the previous study (PMID: 34433568), where the ECD was clearly resolved (nanobody-mediated fixation of the ECD). Additionally, in most of the provided 2D classes density for the ECD is clearly visible (admittedly very cloudy, supporting its flexible nature). Line 135 states that the ECD “is much lower resolution”, however, its not visible at all; therefore, you might want to alter this statement.

As mentioned by the reviewer, the ECD of PepT2 is likely flexible. In our previous study (PMID: 34433568; Fig. S3) we could show that the ECD exists in two conformations: conformation 1 whereby the ECD packs against the nanobody resulting in strong density, and conformation 2 whereby the ECD is rotated ~ 90 degrees away from the nanobody resulting in much weaker density. While we cannot rationalize why the maps presented in our current study favour the latter ECD conformation, it is worth noting that, due to 1) the focus of this study on antibiotic recognition by the TM helices and 2) the overall number of particles going into our reconstructions, our classification scheme focused on improving ligand density and not distinguishing between ECD conformations. Density for the

ECD can be seen in some of the earlier stages of our workflow figures (Fig. S1, S2, S4), though at later stages of the workflow we generally use a higher contour level to minimize appearance of the detergent micelle and highlight side chain density of the TM helices. At lower contour levels density for the ECD is present, albeit weakly and at much lower resolution.

- Line 139: The 562 C-alpha atoms suggest that rather the whole transmembrane domains were used for alignment, i.e., including the loops connecting the TM helices. Please check.

This is correct.

- Figure 4 legend for panel (A) reads wrong ("showing the three orientations overlaid in the binding site") – only two poses are overlaid.

corrected

- Based on the density in Fig. 4B and C, the chlorophenyl group would also fit when rotated by 180° around the bond to the methylisoxazole moiety: Is there any evidence for the current placement?

As the referee likely appreciates, modelling ligands within cryo-EM density (even at moderately high resolution) is tricky and subject to assumptions based on chemistry and best-fit to the available density. In the case of cloxacillin we modelled the chlorine atom in the present position based on the favourable distance to the nearby Arg57 and Lys161 side chains. However, we did consider rotating the chlorophenyl group 180° as it would also fit in the pocket. However, as can be seen in the Coot screenshot, the fit within the density is worse. We thus chose to finalise our model based on the one presented in the manuscript but accept this is just one of several possible positions cloxacillin could adopt.

However, we did consider rotating the chlorophenyl group 180° as it would also fit in the pocket. However, as can be seen in the Coot screenshot, the fit within the density is worse. We thus chose to finalise our model based on the one presented in the manuscript but accept this is just one of several possible positions cloxacillin could adopt.

- What is the r.m.s.d. with the peptide-bound structure of human PepT1 (PDB:7PMX) structure? Please indicate.

We have now included these values in the revised manuscript.

- Sentence in lines 373-375 "...is consistent with the three binding poses..." is confusing: I thought there only two poses observe in cryo-EM structure?

Corrected

- Line 406: not consistent with chapter one stating "cefadroxil displayed an IC50 of 19 μM".

Corrected

- Generally, the purification procedure (lines 501 - 509) is not explained sufficiently: Please provide more details.

done

- Line 577: Please verify if the following statement: "~ 1 e- /Å2 per frame" is correct.

Yes, this is correct.

- The carboxyl group in the schematic compound representation for cefadroxil and amoxicillin in Fig. 3B and D are

deprotonated (as expected at pH 7.5), however, in the corresponding Fig. S5 for cloxacillin the carboxyl group are in a protonated state. Please improve.

Many thanks, this has been corrected.

- Supplementary Information Figs. 1, 2 and 4: Certain 3D classifications yield distributions percentages above 100%, potentially due to rounding errors (e.g., focussed 3D classification in Fig. S1 yields 100.1%) – please correct.

The reviewer is correct in pointing out that class distribution totals will result in percentages of 100 +/- 0.1% due to rounding to the first decimal place. Because we list the overall number of particles belonging to each class in our workflows, we do not feel it statistically necessary to calculate and list class distributions to multiple decimal places.

Minor point: Spelling issues / wording – all corrected.

- Line 75: "binding site" not "biding site"
- Line 77: The reviewer understands "inhibition mechanism", but what is meant by "substrate", respectively, "substrate mechanism" ? Probably, the authors mean "substrate binding and inhibition mechanisms"?
- Sentence in lines 82-86: Quite long sentence and doesn't read nicely, please improve.
- Line 41: delete anaerobic bacteria
- Please make the "i" in the inhibitory constant lowercase, i.e., K_i
- Figure 1c, bar graph: the unit is μM instead of uM
- Line 104: Should read "SURF2R"
- Line 187: remove the space in $\sim 90^\circ$
- Line 225: "...E53xxER..." - unclear what the number 53 stands for. Please provide information to reader.
- Line 543: $3\text{H} - 3$ is currently not superscript
- Line 546: full stop
- Line 516: delete DDM:CHS
- Line 501: Unclear - was 1 x PBS supplemented with 150 mM NaCl or does your PBS include 150 mM NaCl?
- Methods: For consistency, either indicate centrifugation speed as "x g" or simply "g"
- Line 505: reads like a repetition from above.
- Line 554: " μl " instead of "ul"
- Line 372, 372 and 581: spaces are missing
- Line 418: "sp3 hybridized"
- Line 620: should read "non-uniformly refined"
- Line 511: indicate if molar ratio or w/w for the lipids
- Line 553: Hepes should be capitalized for consistency

Reviewer #3 (Remarks to the Author):

The manuscript titled "Structural basis for antibiotic transport and inhibition in PepT2, the mammalian proton-coupled peptide transporter." by Joanne L.Parker, et al combines protein-ligand structural analysis with cryo-em and molecular dynamics (MD) simulations to understand the antibiotics transport and inhibition mechanism in PepT2 transporter. Initially PepT2 is reconstituted into liposomes to compare three different antibiotics in their effect on alanine dipeptide uptake through IC50 experiments. These studies were complemented with counterflow experiments. These showed that cefadroxil and amoxicillin act as ligands while cloxacillin acts as a competitive inhibitor. They solved cryo-em structures of PepT2 bound with three beta-lactam antibiotics - cefadroxil, amoxicillin and cloxacillin, discovered transport mechanism of the former two ligands and provide rationale for cloxacillin functioned as inhibitor instead. When solving the structure bound to cloxacillin, once more the protein structure is close to Apo PepT2. However, different poses are identified for the ligand: one of them establishes interactions similar to the previous ligands, but the possibility of the second binding mode can possibly result in inhibitory behavior. It would be interesting to see based on recent CryoEM methods (e.g Pilar Cosio, Gerhard Hummer) what the population of each pose is. They performed unbiased MD simulations and absolute free energy calculations for both deprotonated and protonated systems. The computations complement their conclusions from experiments for discovering the role of key interactions in transporting cefadroxil and amoxicillin substrates, and help support their hypothesis of protonation mechanism in antibiotic binding and transport. Further analysis of additional substrates and inhibitors provides structural insights in distinguishing substrates and inhibitors.

Standard MD simulations complement the CryoEM by using the CryoEM starting poses and different protonation states. I wonder if constant pH simulations would have been better, since they would allow the protonation/deprotonation of GLU53 and 56 in response to the instantaneous environment while keeping the pH at 7.5. For example, Jana Shen has recently shown that more than the state (protonated/deprotonated) it is the ability to deprotonate which can differ from ligand to ligand.

As the reviewer points out, in theory constant pH MD sounds like the perfect technique that could treat protonation-state changes of E56 together with potential ligand and side-chain rearrangements and ligand conformational changes together. We have recently done this for the Ala-Phe peptide ligand in in PepT2 to show how the pKa of E56 is modulated by the presence of the substrate (Lichtinger Simon M, Parker Joanne L, Newstead Simon, Biggin Philip C (2024) The mechanism of mammalian proton-coupled peptide transporters eLife 13:RP96507 <https://doi.org/10.7554/eLife.96507.2>), although even in that case the rearrangement of the side chain of E56 in response to protonation proved to be the convergence bottleneck.

Unfortunately, achieving convergence of constant pH simulations in situations where slow degrees of freedom (here, the large ligand rearrangement we describe) depend on the protonation states in question is extremely challenging, because at each attempted protonation-state change - whether treated discretely or via a continuous variable - the simulation needs to re-equilibrate to that state (which has a time scale of 100s of ns in the deprotonated -> protonated case, and potentially much longer in the reverse direction, which we have not tested beyond some initial trials). Although we have therefore attempted to run the same protocol as in our other paper for the cefadroxil case, we were not able to converge the simulations and thus did not pursue this avenue further.

Page 15 Table 2, (1) cloxacillin has no error bar compared to the other two, the authors mentioned in the Methods part that only the highest binding affinity replicate is reported for cloxacillin, they should include the rationale behind.

Error bars are reported for stable binding poses of cefadroxil and amoxicillin following extra equilibration and derivation of new restraints. Since cloxacillin did not have a binding pose stable on this equilibration timescale, we could not derive replicates for error bars in the same way. We have clarified this in the manuscript.

They include MBar estimation of all replicates in SI since the mbar estimation for each replicate should have error bar, which are good for checking the consistency of replicates.

Errors reported by mBAR are a quantification of the error in the free energy given the available sampling and processing in a statistically optimal way. These errors are much smaller (in our case, more than one order of magnitude -typically less than 0.04 kcal/mol) than the inter-replicate differences, indicating that the sampling itself, not the free energy estimation is limiting our accuracy here. We therefore see no benefit to the reader in including those.

(2) the authors should provide appropriate reasons for the difference in rankings of binding affinity among cefadroxil, amoxicillin, cloxacillin (in IC50 experiments: cefadroxil > cloxacillin > amoxicillin, while in computations and previously reported Ki values: cefadroxil > amoxicillin > cloxacillin).

There could be several reasons to explain the differences between our IC50 values and those reported previously in the literature, most notable those in Terada et al., 1997 & Luckner et al., 2005. In the present study we calculated IC50 values using reconstituted PepT2 whereas in the previous studies transport was analysed using cell-based assays. The difference in experimental set up could easily account for the differences in ranked order. Most notably, in our liposome assays we are monitoring inhibition of uptake of a physiological peptide (Ala-Phe) whereas in the previous studies they used a non-hydrolysable peptide analogue Gly-Sar (Glycyl-Sarcosine). The difference between our IC50 values and the ΔG results are due to the inability of cloxacillin to adopt a stable pose in the binding site while occupation would result in inhibition regardless of whether this was stable.

Page 21 line 486-487, the authors suggest that binding affinity is not predictive for drug transport. This conclusion is not consistent throughout the manuscript. In page 3 line 96, they suggest higher transport efficiency is consistent

with higher IC50. From the section starting from page 16 line 340, both high efficient substrates - cefadroxil and cefaclor, have high IC50, and inhibitors have low IC50 in general except for the low transport efficiency substrate ampicillin. Apparently the datapoints are not enough to simply draw that conclusion.

We agree with the referee that one cannot assume transport efficiency based on affinity or IC50 measurements alone. In addition, as our study has shown for cloxacillin, ceftibuten, benzyl-penicillin and moxalactam, you need to either assess transport directly using radioactive ligands or counterflow using a reconstituted system.

Minor: (all corrected)

Figure 2B there is a typo in "amxoxicillin"

Page 3 line 75, typo, "biding" should be binding.

Page 6 Figure 2, needs explanations for the scale bar "-12.46-0-16.22" in caption.

Page 10 Figure 4 caption, "...showing the three orientations..." should be "...two orientations..."

Page 13 Figure 5C, need explanations of red/blue colors in caption.

Page 15 line 332-333, the authors suggest that there is no stabilization of binding upon Glu56 protonation, they should include figure/data to support the conclusion. We apologise for missing this call out. The data is shown in Supplementary Fig. 8

Page 17 line 375, "...three binding poses..." should be "...two binding poses..."

Page 19 Figure 7D, the authors should indicate the source of "Inward Facing" structural model. The inward facing model was taken from AlphaFold, this information has been added to the legend.

Reviewer #4 (Remarks to the Author):

I co-reviewed this manuscript with one of the reviewers who provided the listed reports. This is part of the Nature Communications initiative to facilitate training in peer review and to provide appropriate recognition for Early Career Researchers who co-review

We thank the reviewers for the time taken to re-read our revision. Based on their comments we have revised our text further, as detailed below:

Reviewer #2 (Remarks to the Author):

We appreciate the author's efforts to improve the manuscript. While most revisions were properly done in accordance with the Reviewers' requests, the manuscript still requires careful revisions.

- The Reviewer requests a thorough proofreading; there are still several spelling errors and inconsistencies (e.g., IC₅₀ and IC50 (50 subscript, mL and ml, etc.) throughout the manuscript that were not fully corrected. Here a few examples (this list is not exhaustive):

- Line 101 (and elsewhere): The "2" in Lys[Z(NO₂)]-Pro should be lowercase
- Figure 1c, bar graph: the unit is μ M instead of uM
- Line 136: A space is missing in "TMs10-11"
- Line 138: "coulomb" should be capitalized
- Line 198: A letter is missing in "Ly161"
- Lines 383-385: inconsistencies with spaces between numbers and units
- Line 436: Mixture of one and three-letter codes of amino acids in the same sentence, i.e., the one-letter code was used for K161 instead of three-letter code (please stay consistent)
- Line 538: "50KDa" should read 50 kDa

...

- Figure 1c: Unfortunately, not all of my points were addressed in this revision. Please provide information about the confidence intervals for the four representative curves displayed (left panel). Additionally, please include the standard deviations (SD) for the individual data points shown (left panel; the right panel is OK). This will help the reader understand the variation of the individual points at different concentrations. Furthermore and based on the added "n=3" statement in the revised version: Are these technical replicates of the representative IC₅₀ curves? Finally, why not showing a merge of at least three biological replicates here, instead of "representative IC₅₀ curves"? Data seem to be available.

Our IC₅₀ values are calculated from three independent experiments. Each experiment was performed separately on different days. The IC₅₀ is calculated from each experiment and the mean and SD are calculated from these individual experiments. Therefore, there are no SD for each concentration as the data are not merged. Technical replicates for this type of experiment give a value within 99 % and therefore do not show up on the graphs and do not provide useful information in the variation of the calculated IC₅₀ value. The data show on the left-hand side of panel c shows the actual data obtained from one experiment therefore it would be meaningless to put a CI or SD on this type of graph. However, we show the mean and SD in the right-hand panel which demonstrates the spread of data obtained for the calculated IC₅₀ value from the 3 biological, independent replicates. It appears that reviewer 2 has not understood this point so we have added "independent biological experiments" to the figure legend to make this clearer. We have not shown

all the individual graphs as this would make the figure quite messy and hard to interpret.

- Figure 5b: Please consider the points in my previous comment.

As above these are representative curves calculated in the same manner as figure 1.

- Considering the ligand densities: I agree that the correct modelling of ligands in cryo-EM maps at the achieved resolution is tricky and I appreciate the authors statement that the modelled pose is “is just one of several possible positions”. Especially in the case of pose 2 for the cloxacillin-bound structure (Supplementary Figure 6d), considerable portions of the compound are not within the density at the shown threshold. Considering these modelling uncertainties, lines 233-238 should highlight and contain your statement “is just one of several possible positions”.

We have added this information to the main text.

- The arrows used in multiple Figures as hydrogen bond representatives are rather confusing (might indicate donors or acceptors, unclear) and distracting: Please replace with dotted or dashed lines.

We have included the description of the lines and arrows in the figure legend to aid the reader.

- Fig. 2b and c (and also in Suppl. Information): Visualization and measurement of H-bonds is inconsistent. Normally, H-bond distances are indicated and measured between heavy atoms (e.g., Y61 with carbonyl oxygen atom of cefadroxil). However, in the same panel (b in Fig. 2), distances are measured from heavy atoms to hydrogen atoms, e.g., D317 and E622 to ligand H-atoms (instead to ligand heavy atoms, i.e., nitrogen and oxygen atoms). Please carefully adapt concerned Figures and, importantly, check measured distances: It might be needed to perform major changes in distances including in other Figures and in the text, e.g., the D317 to OH-group in Fig. 2d is with 2.12 Å probably too short and not measured in the usual convention.

We have now clarified in the text the distances are all between heavy atoms and removed the hydrogens from these figures. Similarly supplementary Fig 6 has also had the hydrogens removed from the interaction figures.

- Figure 1: The colouring of the compounds is suboptimal – sulphur and nitrogen atoms are barely visible, and the compounds are rather small. Please revise and make use of the empty space within this figure.

We have made these figures larger and standardised the chemical structures between Figs 1 and 5. However the colours are the standard colours used by Chemdraw.

- Figure 3c: Please add the text “EM density” (as in b) for consistency. Furthermore,

please add the membrane boundaries in panel a as illustrated in Fig. 2a and mark the cytoplasm and extracellular sides (also as in Fig. 2a).

Done

- The newly added encircled TM labels (combined with the filled ellipses and amino acid residue labels) overload the respective panels and distract from the actual topic, i.e., the protein-ligand interactions. Please tone them down, e.g., by reducing the sizes of the filled circles and ellipses.

We have reduced the size of the labels.

- Figure 5a: Concentrations for all compounds should be added (i.e., also for Ala-Ala and Cefaclor). Furthermore, "BenzylPenicillin" spells Benzylpenicillin. As a side note: The data points for Ala-Ala are coloured inconsistently.

Added

- Supplementary Figure 6a, left panel: The interaction between Y61 and the sulphur atom is not shown and K161 is not labelled. Please also check panel b for inconsistencies and missing information.

Added

- Supplementary Figure 7 and 8: Please colour plots darker for enhanced contrast and better visualization.

These plots conform to standardised visualise settings.

- Figure 2d: A typical distance between donor and acceptor in salt bridges is about 2.8 Å. The distances between K161 and R57 from the carboxylate of the ligand are both > 4 Å. Therefore, I ask myself if such interactions are significant and no interactions should be drawn in this panel/figure. Furthermore, I suggest to indicate only decimals, e.g., 2.5 Å instead of 2.52 Å.

The referee is correct that salt bridges will be strongest at this distance but nevertheless there will be an electrostatic attraction between the carboxylate group and Lys161 and Arg57 in the transporter. We feel it necessary to draw the readers attention to this attractive electrostatic interaction in these schematics due to the importance of this interaction as highlighted in the MD simulations (Fig. 4). We have amended the text to remove salt bridge and replaced with attractive electrostatic interaction. We have also amended the distances.

- Figure 6c: In the legend it's called "E53xxER", in the panel only "ExxER". Besides of being inconsistent, please provide information about the meaning of "53" to the reader.

done

- Figure 1 legend for panel e: For completeness, please state that the non-transported alanine and cloxacillin were tested as well, as the figure displays SSM traces for these compounds.

done

- Please capitalize "Figure" in all supplementary figure legends.

done

- Line 605: "... 10 e-V ..." - the superscripted minus sign should be removed (the unit for electron-volt is eV)

done